# Kit ligand has a critical role in mouse yolk sac and aorta–gonad–mesonephros hematopoiesis

Emanuele Azzoni[1,*] iD, Vincent Frontera[1], Kathleen E McGrath[2], Joe Harman[1] iD, Joana Carrelha[1,3], Claus Nerlov[1], James Palis[2], Sten Eirik W Jacobsen[1,3,4] & Marella FTR de Bruijn[1,**] iD

## Abstract

Few studies report on the *in vivo* requirement for hematopoietic niche factors in the mammalian embryo. Here, we comprehensively analyze the requirement for Kit ligand (Kitl) in the yolk sac and aorta–gonad–mesonephros (AGM) niche. In-depth analysis of loss-of-function and transgenic reporter mouse models show that Kitl-deficient embryos harbor decreased numbers of yolk sac erythro-myeloid progenitor (EMP) cells, resulting from a proliferation defect following their initial emergence. This EMP defect causes a dramatic decrease in fetal liver erythroid cells prior to the onset of hematopoietic stem cell (HSC)-derived erythropoiesis, and a reduction in tissue-resident macrophages. Pre-HSCs in the AGM require Kitl for survival and maturation, but not proliferation. Although Kitl is expressed widely in all embryonic hematopoietic niches, conditional deletion in endothelial cells recapitulates germ-line loss-of-function phenotypes in AGM and yolk sac, with phenotypic HSCs but not EMPs remaining dependent on endothelial Kitl upon migration to the fetal liver. In conclusion, our data establish Kitl as a critical regulator in the *in vivo* AGM and yolk sac endothelial niche.

**Keywords** AGM; embryo; hematopoiesis; Kit ligand; niche
**Subject Categories** Development & Differentiation; Signal Transduction; Stem Cells

## Introduction

During mouse development, hematopoietic stem and progenitor cells (HSPCs) emerge in asynchronous waves at different anatomical sites [1–3]. The first wave originates in the yolk sac (YS) at E7.25 and generates the primitive erythroid and macrophage progenitors of the murine conceptus [3,4]. From E8.25 onwards, the YS generates a second wave of bi- and multi-potent progenitors of the erythroid, myeloid and lymphoid lineages [3,5,6]. This wave includes the erythro-myeloid progenitors (EMPs) that emerge from the YS endothelium starting from E8.25, become more prevalent by E9.5, and seed the fetal liver (FL) where they maintain hematopoiesis during fetal life [7,8]. Finally, the definitive hematopoietic stem cell (HSC) lineage emerges in the aorta–gonad–mesonephros (AGM) region and vitelline and umbilical arteries (VU), in hematopoietic cell clusters that protrude from the endothelial wall of these arteries [9,10]. In the dorsal aorta, these hematopoietic cell clusters were shown to emerge from the ventral hemogenic endothelium via a continuum of Kit$^+$ pro- and pre-HSCs which after E10.5 mature into definitive HSCs [11–13]. The process by which HSCs and YS EMPs emerge from the hemogenic endothelium is known as endothelial-to-hematopoietic transition (EHT). After E11.5 HSCs colonize the FL, and later the bone marrow (BM) from where they maintain hematopoiesis throughout adult life [14–19].

The extrinsic regulators and niche components that support HSPC generation during embryonic development remain poorly understood. Several soluble factors are reported to regulate HSPC biology in both the developing embryo and adult BM, sometimes with apparently opposing effects [20,21]. However, many functional studies of embryonic/fetal hematopoietic niche factors have been conducted *in vitro*, leaving uncertainty about the physiological role of these factors in HSPC emergence in the YS and AGM [20]. Kit ligand (Kitl; also known as Stem Cell Factor/SCF, or steel factor) is arguably one of the best studied key signaling factors in the adult BM HSC niche, where it binds to and activates the tyrosine kinase receptor Kit on HSPCs, and is responsible for their proliferation and survival [22–24]. In the embryo, Kitl is expressed at hematopoietic sites [25–27], though the cells responsible for its production in the embryonic hematopoietic niche have not been identified. Genetic defects in Kitl/Kit signaling result in late embryonic/perinatal lethality with severe anemia [22,28]. This has been ascribed to an erythroid differentiation block in > E13.5 FL, along with a decrease in FL CFU-S and neonatal HSCs [22,28–32]. While all mouse YS EMPs and emerging AGM HSCs express the Kit receptor [8,9,33,34],

1  MRC Molecular Hematology Unit, MRC Weatherall Institute of Molecular Medicine, Radcliffe Department of Medicine, University of Oxford, Oxford, UK
2  Department of Pediatrics, Center for Pediatric Biomedical Research, University of Rochester Medical Center, Rochester, NY, USA
3  Hematopoietic Stem Cell Laboratory, MRC Weatherall Institute of Molecular Medicine, Radcliffe Department of Medicine, University of Oxford, Oxford, UK
4  Department of Cell and Molecular Biology, Wallenberg Institute for Regenerative Medicine and Department of Medicine, Center for Hematology and Regenerative Medicine, Karolinska Institutet and Karolinska University Hospital, Stockholm, Sweden
   *Corresponding author. Tel: +44 1865 222377; Fax: +44 1865 222424; E-mail: emanuele.azzoni@imm.ox.ac.uk
   **Corresponding author. Tel: +44 1865 222397; Fax: +44 1865 222424; E-mail: marella.debruijn@imm.ox.ac.uk

experiments with receptor-neutralizing antibodies, and the persistence of Kit$^+$ cells in the YS and AGM of embryos with a non-functional Kit receptor, suggested that Kitl/Kit signaling is not required for HSPC emergence in the early embryo [35,36]. More recently, *in vitro* culture data did suggest a role for Kitl in maturation of the AGM HSC lineage [11]. However, the role of Kitl in the YS and AGM hematopoietic niches has not been directly investigated *in vivo*.

Here, we make use of genetic mouse models to investigate the expression of Kitl in the YS and AGM hematopoietic niches during the onset of hematopoiesis and to assess its requirement in YS-derived EMPs, lymphomyeloid progenitors, and the AGM HSC lineage *in vivo*. Our results reveal a previously unrecognized requirement for Kitl in YS-derived EMPs and tissue macrophages, support a role for Kitl in AGM HSC maturation and survival, and firmly establish this cytokine as a critical endothelium-derived YS and AGM hematopoietic niche factor.

# Results

## YS EMPs require Kitl for their normal development

To determine the role of Kitl in YS hematopoiesis, we analyzed Kitl-deficient *Steel* embryos (*Sl/Sl*) that carry a deletion encompassing the *Steel* locus encoding Kitl [24,37–39]. Erythro-myeloid progenitors emerge from the YS endothelium starting from E8.25 and become more prevalent in the YS by E9.5 [7,8]. Erythro-myeloid progenitors are phenotypically defined as Kit$^+$ CD41$^+$ CD16/32$^+$ and comprise a heterogeneous population containing clonogenic progenitors for the erythroid, myeloid, and mixed myeloid/erythroid lineages [8]. EMPs were present in normal frequency and numbers in E9.5 *Sl/Sl* YS (Fig 1A and B) and exhibited normal clonogenic potential at both E8.5 (Fig EV1A) and E9.5 (Fig 1C, left panel). By E11.5, however, *Sl/Sl* YS EMPs were significantly reduced compared to wild-type littermates, both phenotypically (Fig 1A and D) and functionally (Fig 1C, right panel).

In contrast, primitive erythroblasts were not affected in *Sl/Sl* embryos (Fig EV1B and C), in accordance with the reported normal development of this lineage in embryos with severely reduced levels of Kitl [31]. We next assessed whether defects in proliferation and/or survival could underlie the YS EMP defect, as Kitl is known to control cell cycle and/or promote survival of other HSPCs [23,40–42]. Analysis of phospho-histone H3 expression (pHH3, a marker of mitotic cells; Fig 1E) and BrdU incorporation (Fig 1F) showed that proliferation of YS EMPs was reduced, starting with an approximately twofold decrease already at E9.5, and still apparent at E11.5. Apoptosis, on the other hand, was not significantly affected in *Sl/Sl* EMPs (Fig EV1D). Taken together, these data demonstrate a previously unrecognized requirement for Kitl in YS EMP proliferation and the normal generation of the YS EMP pool.

## The onset of *Steel* FL anemia precedes HSC-derived FL hematopoiesis and is due to the defect in YS-derived EMPs

YS-derived EMPs colonize the FL and contribute to hematopoiesis at this site [8,43]. Already at E11.5, the number of Kit$^+$ CD41$^+$ CD16/32$^+$ EMPs and EMP-derived Kit$^+$ CD41$^-$ CD71$^+$ CD44$^+$ early erythroid progenitors [44] was dramatically reduced (3-fold and 30-fold, respectively), along with an overall decrease in E11.5 *Sl/Sl* FL cellularity (Fig 2A–C) and clonogenic progenitors (Fig 2D). By E12.5, total FL cellularity and CFU-E numbers were further reduced in *Sl/Sl* mutants (Fig 2E). Imaging flow cytometry revealed a severe reduction in the number of terminally differentiating erythroid lineage cells at this fetal stage, both at the level of the Kit$^+$ proerythroblast (ProE, ninefold decreased compared to wild type) and the downstream Kit$^{lo/-}$ erythroid precursors (threefold reduced compared to wild type; Fig 2F and G). Of note, the relative abundance of the different types of Kit$^{lo/-}$ erythroid precursors in E12.5 *Sl/Sl* FL was normal, indicating that loss of Kitl dramatically affects the overall extent of erythropoiesis, but not the capacity of the cells to undergo terminal maturation. Importantly, the observed erythroid defects in E11.5/12.5 *Sl/Sl* FL are too early to be explained by a

---

**Figure 1. YS *Sl/Sl* EMPs fail to reach normal numbers after their initial generation.**

A    Number of EMPs per embryo equivalent (e.e.) of wild type and *Sl/Sl* E9.5 and E11.5 YS, determined by flow cytometry (panels in B,D). E9.5 Kit$^+$ CD41$^+$ CD16/32$^+$ EMP numbers are the mean ± SD from four wild type and three *Sl/Sl* biological replicates, with each replicate consisting of single or two pooled YS of the same genotype. Total number of embryos analyzed: 7 +/+ (14–23 sp), 5 *Sl/Sl* (17–25 sp). E11.5 Kit$^+$ CD41$^+$ EMP numbers are the mean ± SD of 5 +/+ and 6 *Sl/Sl* YS analyzed individually over two independent experiments. Total live cells per YS: 1.7 ± 0.3 × 10$^5$ (wt), 1.7 ± 0.2 × 10$^5$ (*Sl/Sl*) at E9.5; 4.2 ± 1.4 × 10$^5$ (wt), 4.0 ± 2.2 × 10$^5$ (*Sl/Sl*) at E11.5.

B    Example of flow cytometry analysis of E9.5 YS EMPs quantified in panel (A). Analyses were performed on Ter119$^-$ live cells.

C    Number of CFU-C in wild type and *Sl/Sl* E9.5 and E11.5 YS. E9.5 data (mean ± SD) are from three wild type and two *Sl/Sl* biological replicates plated in duplicate, with each replicate consisting of single or two pooled YS of the same genotype. Total number of analyzed embryos: 5 +/+ (17–23 sp), 3 *Sl/Sl* (17–25 sp) over two independent experiments. For E11.5, data are from four wild type and six *Sl/Sl* biological replicates plated in duplicate, with each replicate consisting of single or two pooled YSs of the same genotype. Total number of embryos analyzed: 6 +/+, 7 *Sl/Sl* over four independent experiments. GEMM: granulocyte, erythroid, monocyte/macrophage, megakaryocyte; G/M/GM: granulocyte, monocyte/macrophage; Ery: erythroid.

D    Example of flow cytometry analysis of E11.5 YS EMPs quantified in panel (A). Analyses were performed on Ter119$^-$ live cells.

E    Confocal WM-IF analysis of CD31$^+$ pHH3$^+$ Kit$^+$ proliferating EMPs (arrowheads) in E9.5 wild type and *Sl/Sl* YS. A single 2.5-μm-thick Z section is shown. For each YS, CD31$^+$ Kit$^+$ pHH3$^+$ cells were counted in 3 to 5 fields using a 25× objective and the percentage of pHH3$^+$ proliferating EMPs was calculated. Bar graphs represent the mean ± SD from three wild type (21–24 sp) and three *Sl/Sl* (22–23 sp) YS. Scale bars: 50 μm.

F    Cell cycle analysis of wild type and *Sl/Sl* E11.5 YS EMPs. The percentage of EMPs (Ter119$^-$ Kit$^+$ CD41$^+$) in G0/G1 or S/G2/M was determined by flow cytometry on the basis of BrdU and 7-AAD incorporation. a: apoptotic (percentage not shown). Embryos were analyzed individually. *N* = 6 (+/+), *N* = 5 (*Sl/Sl*) over three independent experiments. E11.5 embryos were staged according to tail somite counts as described in ref. [76] and were as follows: 12–17 (+/+); 12–17 (*Sl/Sl*). Bar graphs show mean ± SD, and FACS plots representative results.

Data information: *P* < 0.05 (unpaired two-tailed Student's *t*-test).

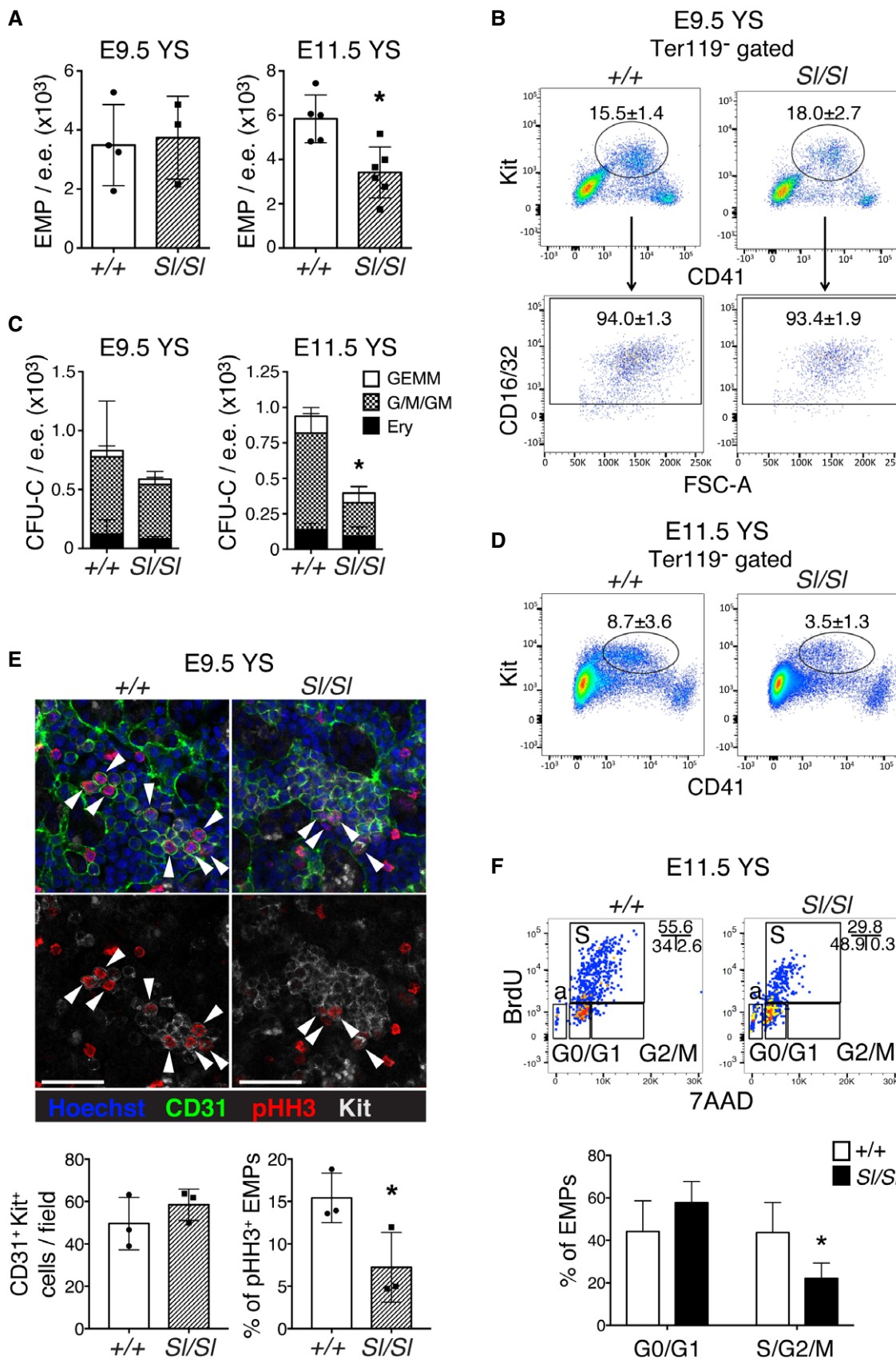

**Figure 1.**

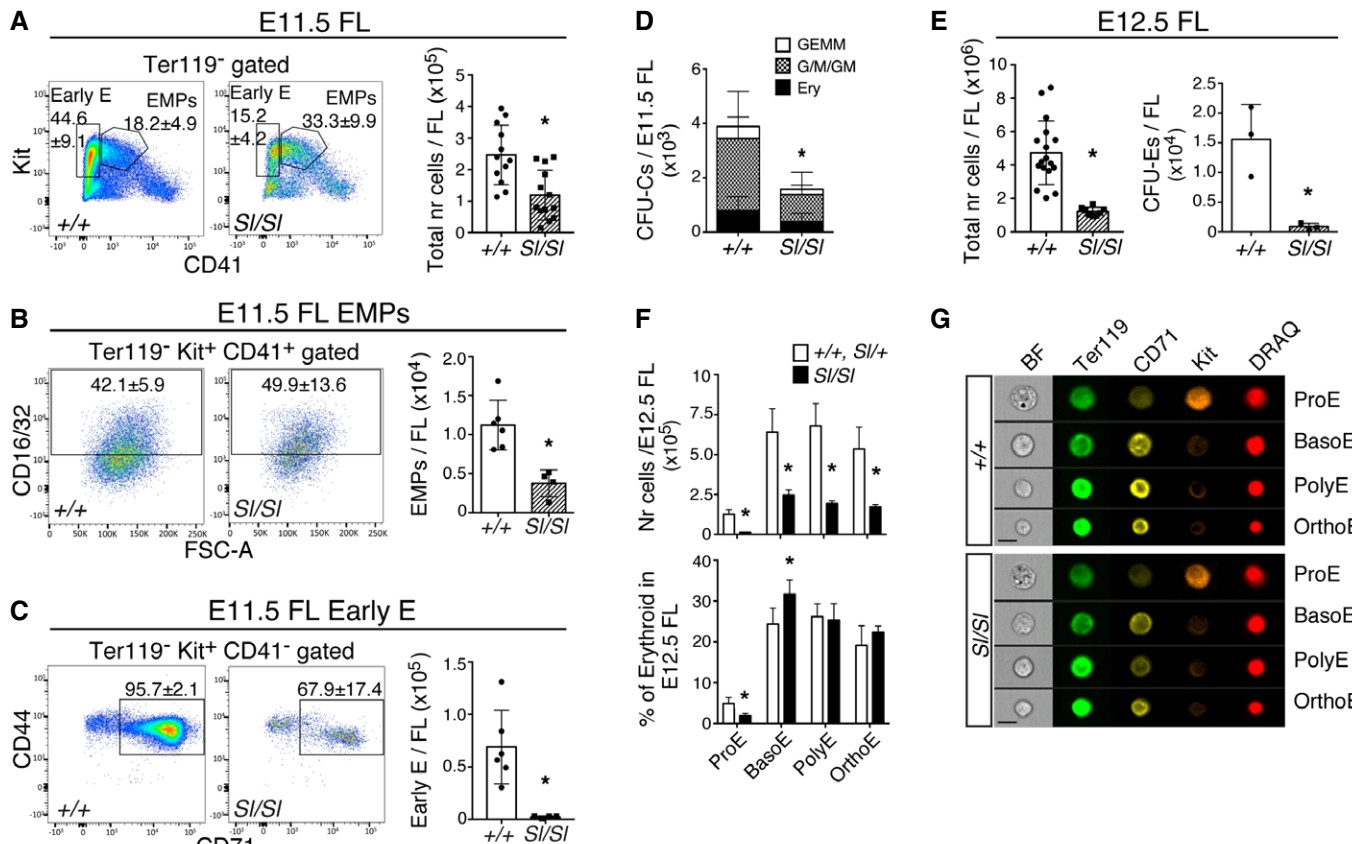

**Figure 2.  Kitl is required for the early FL expansion of EMPs and their progeny.**

A–C   Flow cytometry analysis of EMPs (Ter119⁻ Kit⁺ CD41⁺ CD16/32⁺) and early erythroid progenitors (Early E: Ter119⁻ Kit⁺ CD41⁻ CD71⁺ CD44⁺) in wild type and *Sl/Sl*
       E11.5 FL. EMP and early E data are the mean (± SD) of six wild type and four *Sl/Sl* biological replicates, with a total of 10 wild type and 6 *Sl/Sl* embryos analyzed in
       pools of 1–3 FL per genotype. Bar graphs on the right show corresponding absolute cell numbers (nr) per FL. Total FL cell counts are the mean (± SD) of 12 wild
       type and 13 *Sl/Sl* embryos. Tail somite range: 10–16 (wild type), 12–15 (*Sl/Sl*). Note that although EMP frequency in *Sl/Sl* FL is increased, their absolute number is
       severely decreased.
D     CFU-C numbers in wild type and *Sl/Sl* E11.5 FL. Data are the mean (± SD) of 10 wild type and 10 *Sl/Sl* biological replicates plated in duplicate, with each replicate
       consisting of cell from 1 to 2 FLs. A total of 12 wild type and 11 *Sl/Sl* FL were plated over six independent experiments. GEMM: granulocyte, erythroid,
       monocyte/macrophage, megakaryocyte; G/M/GM: granulocyte, monocyte/macrophage; Ery: erythroid.
E     Left: Total cell numbers (nr) of E12.5 wild type (*n* = 7) and *Sl/Sl* (*n* = 8) FL. Right: CFU-E numbers per wild type and *Sl/Sl* E12.5 FL. Data are the mean (± SD) of
       three biological replicates analyzed over two independent experiments. Each biological replicate consisted of 1–2 FL samples pooled according to genotype and
       was plated in duplicate into a methylcellulose based medium containing EPO; colonies were scored after 2 days. Total number of FLs analyzed: four wild type and
       six *Sl/Sl*.
F     Quantification of erythroid lineage cells in E12.5 wild type and *Sl/Sl* FL. Absolute cell counts (top) and percentages (bottom) of proerythroblasts (ProE), basophilic
       erythroblasts (BasoE), polychromatic erythroblasts (PolyE), and orthochromatic erythroblasts (OrthoE) were determined by imaging flow cytometry analysis. The
       number (nr) of cells per FL was based on total cellularity and their relative proportions. Data are the mean (± SD) of three biological replicates for each genotype,
       with replicates consisting of single or two pooled FL samples. A total of four wild type, five *Sl/+*, and six *Sl/Sl* FLs were analyzed.
G     Representative examples of maturing erythroblast series in wild type and *Sl/Sl* E12.5 FL, as identified by imaging flow cytometry. Scale bars: 10 μm.

Data information: **P* < 0.05 (unpaired two-tailed Student's *t*-test).

defect in hematopoiesis downstream of FL HSCs, as very few long-term repopulating (LTR)-HSCs are present in the FL before E13.5 [19,45].

The observed FL phenotype could be due to a decreased influx of YS-derived EMPs into the FL and/or a decrease in proliferation of erythroid lineage cells. In support of the former, the number of circulating EMPs showed a decreased trend in *Sl/Sl* peripheral blood (PB) at E9.75, the time at which FL colonization commences (Figs 3A, top and EV1E and F), and was significantly reduced by E11.5 (Figs 3A, bottom and EV1F). While Kit⁺ cells seeded the late

E9 (27–29 sp) FL primordium of wild type and *Sl/Sl* embryos alike (Fig 3B, top), indicative of a normal temporal onset of FL colonization [46], the *Sl/Sl* mutant FL contained fewer Kit⁺ cells at E10.5 (34–36 sp) (Fig 3B, bottom), suggesting a reduced total influx of EMPs. In addition, a defect in progenitor cell proliferation *in situ* may contribute to the reduction in Kit⁺ cells. Indeed, the absence of Kitl negatively affected proliferation among *Sl/Sl* Kit⁺ cells in the E10.5 FL (Fig 3C), and both EMPs and early erythroid progenitors of the E11.5 *Sl/Sl* FL showed a decrease in proliferation and survival (Fig 3D–G). Thus, our data indicate that the FL anemia of *Steel*

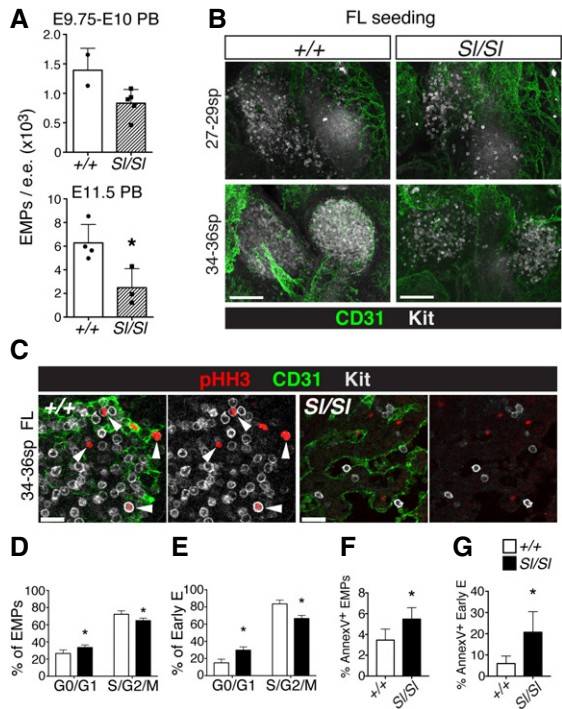

**Figure 3. In the absence of Kitl, the influx and proliferation of EMPs in the early FL are reduced.**

A   Number of EMPs in peripheral blood (PB) of wild type and *Sl/Sl* midgestation embryos, calculated from the percentage of PB EMPs (Ter119$^-$ Kit$^+$ CD41$^+$ CD16/32$^+$) determined by flow cytometry (Fig EV1E and F) and the total cell count (4.2 ± 1.6 × 10$^5$ (wt), 3.1 ± 1.1 × 10$^5$ (*Sl/Sl*) at E9.75; 2.6 ± 0.8 × 10$^6$ (wt), 2.3 ± 1.4 × 10$^6$ (*Sl/Sl*) at E11.5). For E9.75-E10, PB from two wild type (28–30 sp) and five *Sl/Sl* (27–31) individual embryos was analyzed. For E11.5, PB from 1 to 3 embryos was pooled and six wild type and four *Sl/Sl* biological replicates were analyzed, with 10 wild type and six *Sl/Sl* embryos analyzed in total. Graphs show the mean (± SD) of PB EMPs per embryo equivalent (e.e.).

B   Representative confocal whole-mount immunofluorescence images of wild type and *Sl/Sl* FL at the onset of hematopoietic colonization. Kit$^+$ EMPs are shown in white, CD31$^+$ vasculature in green. Maximum intensity 3D projections from 350-μm-thick Z-stacks are shown. Scale bars: 100 μm. Number of embryos analyzed: three wild type and three *Sl/Sl* for E9.75 (27–29 sp), seven wild type, and eight (*Sl/Sl*) for E10.5 (34–36 sp).

C   Confocal whole-mount immunofluorescence analysis of proliferating hematopoietic progenitors in wild type and *Sl/Sl* E10.5 (34–36 sp) FL. Very few CD31$^+$ pHH3$^+$ Kit$^+$ proliferating hematopoietic cells (arrowheads) were detected in the *Sl/Sl* FL. A single 2.5-μm-thick slice is shown. Scale bars: 20 μm. A total of seven wild type and eight *Sl/Sl* embryos were analyzed.

D, E   Percentage of cycling Ter119$^-$ Kit$^+$ CD41$^+$ CD16/32$^+$ EMPs and Ter119$^-$ Kit$^+$ CD41$^-$ CD71$^+$ CD44$^+$ early erythroid (Early E) cells in the E11.5 FL, determined by flow cytometric analysis of BrdU and 7-AAD incorporation. Data are the mean (± SD) of six wild type and five *Sl/Sl* FLs analyzed individually over three independent experiments. Tail somite range: 12–17 (+/+); 12–17 (*Sl/Sl*).

F, G   Percentage of early apoptotic EMPs and Early E in wild type and *Sl/Sl* E11.5 FL, determined by flow cytometric analysis of Annexin-V staining among 7-AAD$^-$ Ter119$^-$ CD41$^+$ Kit$^+$ CD16/32$^+$ EMPs and 7-AAD$^-$ Ter119$^-$ Kit$^+$ CD41$^-$ CD71$^+$ CD44$^+$ Early E. Data are the mean (± SD) of nine wild type and seven *Sl/Sl* biological replicates analyzed over three independent experiments, with each replicate consisting of 1-3 FLs of identical genotypes. A total number of 13 wild type and 9 *Sl/Sl* embryos were analyzed. Tail somite range: 11–17 for both genotypes.

Data information: *$P$ < 0.05 (unpaired two-tailed Student's *t*-test).

embryos is due to a defect in YS-derived EMPs, with reduced colonization of the FL and a reduced proliferation and survival of FL erythroid cells contributing to the phenotype.

### Tissue macrophages are depleted in the absence of Kitl

We next examined whether the generation of tissue macrophages was affected by a lack of Kitl, as YS-derived EMPs were recently shown to contribute to the pool of fetal-derived tissue macrophages during embryogenesis [43,47]. Macrophages first appear in the brain and liver, and later in the developing organs of the fetus where they undergo their final specification [47]. Flow cytometric analysis of organs and tissues dissected from E14.5 *Sl/Sl* embryos showed that brain macrophages were unaffected (Fig 4A). In contrast, in the E14.5 FL both the F4/80$^{hi}$ CD11b$^{lo}$ and F4/80$^{lo}$ CD11b$^{hi}$ macrophage subsets [48,49] were significantly reduced in the absence of Kitl (Fig 4B), as were the macrophages of the E14.5 *Sl/Sl* skin (Fig 4A, Appendix Fig S1A). In the limb buds and lungs, only the F4/80$^{lo}$ CD11b$^{hi}$ subsets were significantly decreased (Fig 4A). Whole-mount immunofluorescence confirmed depletion of CD11b$^+$ cells in skin and limb buds, while F4/80$^+$ cells appeared reduced only in skin, in line with flow cytometry results (Appendix Fig S1B). Earlier in development, we observed a 50% decrease in F4/80$^+$ CD11b$^+$ macrophages in the midgestation AGM region of *Sl/Sl* embryos (Fig 4C–E), but not in yolk sac or fetal liver (Appendix Fig S1C). Importantly, macrophages in hematopoietic cell clusters attached to the wall of the dorsal aorta were present in normal numbers (Fig 4E and F). In conclusion, defective EMP expansion in the absence of Kitl results in a depletion of fetal macrophages in all tissues analyzed, with exception of the brain and aortic cell clusters.

### LMPPs are decreased in the absence of Kitl

Next, we assessed whether Kit-expressing lymphomyeloid immune-restricted progenitors (LMPPs) were affected by a loss of Kitl. Similarly to EMPs, these cells emerge in the YS and colonize the FL prior to emergence of definitive HSCs [6]. Although not statistically significant, the total number of LMPPs in E11.5 *Sl/Sl* FL was decreased (Fig EV2A and B). The contrasting increased frequency of LMPPs in FL is most likely secondary to the severe reduction in erythroid cells described above. In the *Sl/Sl* peripheral blood, LMPPs were decreased in both frequency and number (Fig EV2C). These results suggest that LMPPs also depend on Kitl for their expansion in the YS and/or FL.

### Loss of Kitl negatively affects the AGM HSC lineage *in vivo*

To investigate whether the emerging HSC lineage depends on Kitl *in vivo*, we performed morphological, phenotypic, and functional analyses of the midgestation AGM. In the absence of Kitl, the number of Kit$^+$ cells per E10.5 dorsal aorta was significantly decreased (Fig 5A). Quantification of Kit$^+$ hematopoietic cell clusters (defined as ≥ 2 cells) showed a decrease from 88.4 ± 19.7 to 65.6 ± 14.4 clusters in the wild type versus *Sl/Sl* aorta, respectively ($P$ = 0.03; 7 embryos analyzed per genotype). Detailed analysis of cluster size and position along the wall of the aorta revealed that specifically large clusters (> 10 cells) associated with the ventral wall of the aorta were reduced in number, while smaller clusters

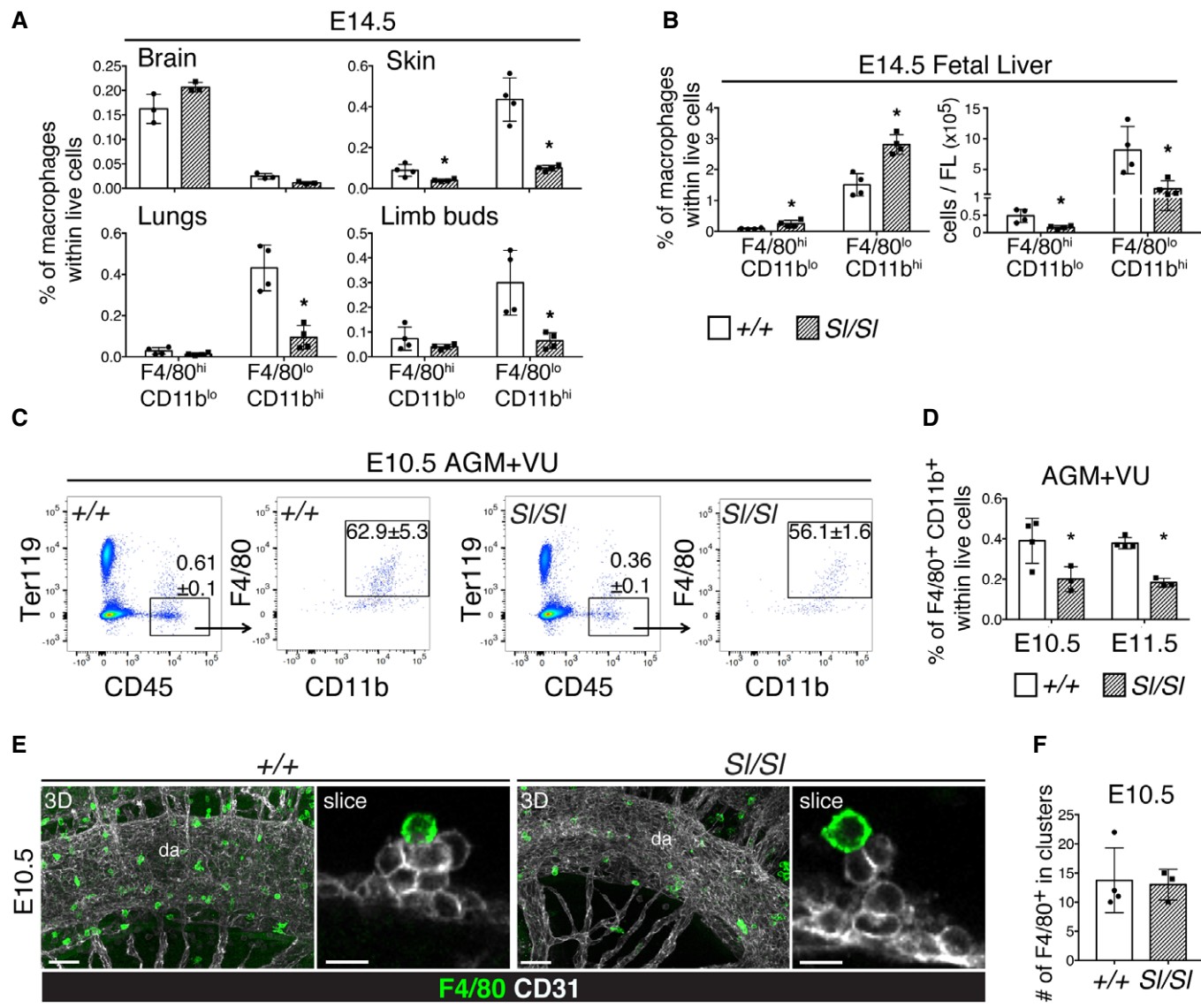

**Figure 4. Macrophages are depleted in *Sl/Sl* mutant embryos.**

A   Percentages of macrophage populations in wild type and *Sl/Sl* E14.5 brain, skin (dissected from the back), lungs, and limb buds, as determined by flow cytometry. Cells were gated as Ter119⁻ CD45⁺ and further gated into F4/80^hi CD11b^lo or F4/80^lo CD11b^hi populations. Representative dot plots of skin analysis are shown in Appendix Fig S1A. Data are the mean (± SD) of four biological replicates for either genotype, with each replicate consisting of individual or pooled samples (from up to three embryos). A total of eight wild type and five *Sl/Sl* embryos were analyzed over two independent experiments.

B   Percentage and absolute numbers of tissue macrophages per E14.5 FL. Data are mean (± SD). Number of embryos analyzed: as in (A).

C   Representative flow cytometry analysis of macrophages in wild type and *Sl/Sl* E10.5 AGM+VU. Cells were gated as shown, with percentages in individual gates as indicated. Data are the mean (± SD) of four wild type (32–35 sp) or three *Sl/Sl* (34–35 sp) biological replicates, analyzed over two independent experiments. Embryos were analyzed individually.

D   Quantification of flow cytometry analysis in (C), showing percentage of F4/80⁺CD11b⁺ macrophages within live cells. E11.5 data are the mean (± SD) of four wild type (10–13 tail somites) or three *Sl/Sl* (10–13 tail somites) biological replicates, analyzed over three independent experiments. Embryos were analyzed individually.

E   Confocal whole-mount immunofluorescence analysis of macrophages in wild type and *Sl/Sl* E10.5 AGM+VU. 3D panels show maximum intensity projections from 400-μm-thick Z-stacks. Slice panels show single 2.5-μm-thick Z-slices of clusters in the dorsal aorta (da). A total of four wild type (31–34 sp) and three *Sl/Sl* (31–33 sp) embryos were analyzed. Scale bars: 50 μm (3D), 10 μm (slice).

F   Quantification of whole-mount immunofluorescence in (E), showing absolute numbers (mean ± SD) of F4/80⁺ macrophages within hematopoietic clusters in the dorsal aorta.

Data information: *P < 0.05 (unpaired two-tailed Student's t-test).

and hematopoietic cell clusters at the dorsal aspect appeared unaffected by the loss of Kitl (Fig 5B), supporting a role for Kitl in AGM HSCs *in vivo*. Circulating Kit⁺ cells, located in the lumen of the aorta, were also decreased (Fig 5B). At the molecular level, RNA-Seq of E11.5 wild type and *Sl/Sl* cluster cells (Ter119⁻ VE-Cadherin⁺ Kit⁺) showed an upregulation of endothelial-associated

**Figure 5.**

genes and a downregulation of hematopoietic genes in the mutant (Fig EV3A and B). Endothelial cells analyzed in parallel as a control showed only few changes (Fig EV3C and D), consistent with their low Kit expression (Fig EV3E). In accordance with decreased hematopoietic gene expression in the AGM clusters, phenotypic pre-HSC type II/HSCs (Ter119⁻ VE-Cadherin⁺ Kit⁺ CD41⁺ CD45⁺) were significantly reduced in the E10.5 and E11.5 Sl/Sl AGM+VU. Pre-HSC type I (Ter119⁻ VE-Cadherin⁺ Kit⁺ CD41⁺ CD45⁻), in contrast, appeared unaffected (Fig 5C and D). Functionally, we observed a marked reduction in clonogenic progenitors in the E11.5 Sl/Sl AGM+VU (Fig 5E), as well as an impaired potential for long-term multi-lineage hematopoietic reconstitution upon transplantation of mutant AGM+VU cells into adult irradiated recipients

(Fig 5F; Appendix Fig S2A–G). Of note, recipients showed a normal relative contribution of Sl/Sl-derived cells to the myeloid and lymphoid lineages (Appendix Fig S2H–L). Our data indicate that in a Kitl-deficient AGM niche overall HSC activity, but not their relative lymphomyeloid lineage output, is negatively affected.

The smaller size of the Sl/Sl aortic clusters suggested a decrease in proliferation and/or increased apoptosis in emerging AGM HSPCs. Whole-mount immunofluorescence analysis showed that the number of proliferating pHH3⁺ CD31⁺ Kit⁺ cells in the E10.5 Sl/Sl dorsal aorta was down twofold compared to wild type (Fig 5G), while the 3number of apoptotic cCasp3⁺ CD31⁺ Kit⁺ cells was increased threefold in the mutant (Fig 5H). Interestingly, no decrease in proliferation was seen among phenotypic Sl/Sl pre-HSC type II/HSCs

**Figure 5.  Kitl is required for AGM (pre-)HSC maturation and survival *in vivo*.**

A　Confocal whole-mount immunofluorescence analysis (WM-IF) of Kit+ cells in wild type and *Sl/Sl* E10.5 AGM (32–36 sp). Images are 3D projections from 400-μm-thick *Z*-stacks, and are representative of 14 wild type and 11 *Sl/Sl* embryos analyzed. Total numbers of Kit+ cells (white) in the dorsal aorta were quantified and showed a significant reduction in *Sl/Sl* compared to wild type (*P* = 0.008). CD31+ vasculature is shown in green. Arrowheads: hematopoietic cell clusters. Scale bars: 100 μm.

B　Quantification of WM-IF in (A), showing absolute numbers (± SD) of clusters of indicated size and separately shown relative to location. A total of seven wild type and seven *Sl/Sl* embryos were counted.

C　Flow cytometry analysis of the HSC lineage in wild type and *Sl/Sl* E11.5 AGM+VU. Cells were Ter119− VE-Cadherin+ Kit+ gated, and percentages of cells within the pre-HSC type I gate (red) and the pre-HSC type II/HSC gate (green) are shown. Percentages are the mean (± SD) of 10 wild type and 9 *Sl/Sl* biological replicates, with each replicate consisting of 1–3 pooled AGM+VU. A total of 15 wild type and 11 *Sl/Sl* embryos were analyzed over five independent experiments.

D　Absolute numbers (± SD) of E10.5 and E11.5 pre-HSC type I (Ter119− VE-Cadherin+ Kit+ CD41+ CD45−) and pre-HSC type II/HSC (Ter119− VE-Cadherin+ Kit+ CD41+ CD45+), as determined by flow cytometry in (C). Number of E11.5 embryos analyzed: as in (A). For E10.5, five biological replicates were analyzed for either genotype, with each replicate consisting of 1–4 pooled AGM+VU. A total of 12 wild type (31–35 sp) and 9 *Sl/Sl* (30–34 sp) embryos were analyzed over four independent experiments. Total live cells per AGM+VU: $2.8 \pm 1.0 \times 10^5$ (wt), $2.3 \pm 0.6 \times 10^5$ (*Sl/Sl*) at E10.5; $3.9 \pm 1.1 \times 10^5$ (wt), $3.8 \pm 1.4 \times 10^5$ (*Sl/Sl*) at E11.5.

E　CFU-C numbers per wild type and *Sl/Sl* E11.5 AGM+VU. Counts are the mean (± SD) of four wild type and six *Sl/Sl* biological replicates plated in duplicate. Biological replicates consisted of cells from individual or two pooled AGM+VU. A total of six wild type and seven *Sl/Sl* AGM+VU embryos were analyzed over four independent experiments. GEMM: granulocyte, erythroid, monocyte/macrophage, megakaryocyte; G/M/GM: granulocyte, monocyte/macrophage; Ery: erythroid.

F　Analysis of long-term multi-lineage HSC potential in E11.5 *Sl/Sl* AGM+VU. Irradiated CD45.1 syngeneic mice were transplanted with 1 e.e. of wild type, *Sl/+*, or *Sl/Sl* CD45.2+ AGM+VU cells. PB chimerism is represented as the percentage of donor CD45.2+ cells among total CD45+ cells, 16 weeks after transplant. A total of 14 recipients were transplanted with wild type or *Sl/+* cells and 8 with *Sl/Sl* cells, over four independent experiments. Wild type is represented with a circle, *Sl/+* a triangle, and *Sl/Sl* with a square. Tail somite range: 9–17 (+/+, *Sl/+*); 10–17 (*Sl/Sl*). \**P* = 0.027, one-tailed Mann–Whitney *U*-test; *P* = 0.055, two-tailed Mann–Whitney *U*-test.

G　Confocal WM-IF analysis of pHH3 expression in Kit+ cells of wild type and *Sl/Sl* E10.5 dorsal aorta (33–36 sp). Images are 2.5-μm-thick single longitudinal *Z*-slices; dorsal up. Arrowheads indicate examples of proliferating CD31+ pHH3+ Kit+ hematopoietic cells (magnified in insets); the total number of these cells per dorsal aorta was quantified and showed reduced proliferation in the mutant (*P* = 0.01). A total of four embryos per genotype were analyzed. Scale bars: 50μm.

H　Confocal WM-IF analysis of cCasp3 expression in Kit+ cells of wild type and *Sl/Sl* E10.5 dorsal aorta (33–36 sp). 2.5-μm-thick single longitudinal *Z*-slices; dorsal up. A higher number of Kit+ cCasp3+ apoptotic cluster cells were detected in the *Sl/Sl* dorsal aorta (*P* = 0.04). Arrowheads indicate a CD31+ Kit+ cCasp3− hematopoietic cluster for wild type and an apoptotic CD31+ Kit+ cCasp3+ hematopoietic cell for *Sl/Sl* (both magnified in insets). A total of four wild type and three *Sl/Sl* embryos were analyzed. Scale bars: 50 μm.

I　Analysis of apoptosis among pre-HSC type II/HSCs in wild type and *Sl/Sl* E11.5 AGM. Ter119− VE-Cadherin+ Kit+ CD41+ CD45+ cells were analyzed by flow cytometry for Annexin-V binding to detect early apoptotic cells. Percentages are the mean (± SD) of nine wild type and seven Sl/Sl biological replicates analyzed over three independent experiments. Individual replicates consisted of 1–3 AGMs. A total of 13 wild type and 9 *Sl/Sl* embryos (11–17 tail sp) were analyzed. The percentage of apoptotic pre-HSC type II/HSCs in the mutant AGM was significantly increased (*P* = 0.04).

J　Multiplex Fluidigm qRT–PCR analysis of pre-HSC type I and II, sorted as shown in (C) from E11.5 wild type (*n* = 4 biological replicates; total number of embryos: 9; 11–17 tail somites) and *Sl/Sl* (*n* = 5 biological replicates; total number of embryos: 9; 11–17 tail somites). Data are mean (± SD).

Data information: \**P* < 0.05 (unpaired two-tailed Student's *t*-test in all panels except F).

(Ter119− VE-Cadherin+ Kit+ CD41+ CD45+), when assayed directly at E11.5 by *in vivo* BrdU labeling (Appendix Fig S2M). In contrast, these cells did show an increase in Annexin-V+ apoptotic cells (Fig 5I), along with reduced expression of the anti-apoptotic gene *Bcl2* (Fig 5J). Conversely, the pro-apoptotic gene *Bax* was upregulated in pre-HSC type I and showed an increased trend among HSCs/pre-HSC type II (Fig 5J). Altogether, these data support a role for Kitl in promoting the survival of the AGM HSC lineage *in vivo*.

**Figure 6.  Spatiotemporal expression of *Kitl*-tdTomato at pre-liver sites of hematopoiesis.**

A　Confocal whole-mount immunofluorescence (WM-IF) analysis of *Kitl*-tdTomato expression in the E8.5 (6–7 sp) conceptus (maximum intensity 3D projection from a 500-μm-thick *Z*-stack). *Kitl*-tdTomato expression (red) is seen in the YS, allantois (a; from which the umbilical artery derives), and para-aortic splanchnopleura (PAS) of *Kitl*-tdTomato::23GFP transgenic concepti (the embryo proper, from which the head has been removed, is outlined with a purple dashed line). The 23GFP transgene (green) labels the YS blood islands (bi; yellow dashed line); this GFP reporter is transcribed from a heterologous hsp68 promoter under the spatiotemporal control of the *Runx1* hematopoietic +23 enhancer, resulting in GFP expression in all primitive erythroid(-fated) cells, hemogenic endothelium, and emerging HSPCs [2,74]. Endothelial cells (CD31+) are in white. Arrowheads indicate the paired aortae (pa). Purple boxed area is magnified in (B); yellow boxed area is magnified in (C). Scale bar: 100 μm. *N* = 2 embryos analyzed.

B　Single 2.5-μm-thick *Z*-slice from the region highlighted in the purple box in (A). Arrowheads indicate *Kitl*-tdTomato+ CD31+ endothelial cells in one of the paired aortae (pa). Scale bar: 25 μm.

C　Single 2.5-μm-thick *Z*-slice from the region highlighted in the yellow box in (A), showing *Kitl*-tdTomato expression in the E8.5 YS BI. Arrowheads indicate *Kitl*-tdTomato+ CD31+ endothelial cells; the arrow indicates a *Kitl*-tdTomato+ perivascular cell. Scale bar: 25 μm.

D　Single 2.5-μm-thick *Z*-slice of 10.5 (35 sp) YS (WM-IF) showing *Kitl*-tdTomato (red), Kit (white), and CD31 (green) expression. Arrowheads indicate *Kitl*-tdTomato+ CD31+ endothelial cells. *N* = 3 embryos analyzed. Scale bar: 50 μm.

E　Confocal WM-IF analysis of *Kitl*-tdTomato expression (red) in a E10.5 (36 sp) AGM region. Top panels (3D view) show a maximum intensity 3D projection from a 400-μm-thick *Z*-stack. Bottom panels (slice view) show a single 2.5-μm-thick *Z*-slice of the dorsal aorta, from the boxed region in the top panels. CD31 (vasculature; green); Kit (hematopoietic progenitors; white). Scale bars: 100 μm (top); 25 μm (bottom). fl: fetal liver; va: vitelline artery plexus; da: dorsal aorta; fp: floor plate; nt: neural tube. *N* = 6 embryos analyzed.

F　Confocal immunofluorescence analysis of cryosections through the AGM region of E10.5 (34–36 sp) *Kitl*-tdTomato transgenic embryos. Arrowheads indicate *Kitl*-tdTomato+ CD31+ endothelial cells (top), *Kitl*-tdTomato+ PDGFR-β+ (middle), or *Kitl*-tdTomato+ α-SMA+ (bottom) perivascular smooth muscle cells. Scale bars: 25 μm or 50 μm (PDGFR-β panel). *N* > 6 embryos analyzed.

G　Confocal immunofluorescence analysis of *Kitl*-tdTomato expression in the E10.5 FL. A cryosection through a E10.5 (35–36 sp) *Kitl*-tdTomato transgenic FL is shown. Arrowheads indicate *Kitl*-tdTomato+ CD31+ Kit− endothelial cells. Asterisks indicate *Kitl*-tdTomato+ CD31− Kit− mesenchymal cells or hepatoblasts. Note proximity of *Kitl*-tdTomato+ cells and Kit+ hematopoietic cells. Scale bars: 50 μm (left panel), 25 μm (magnified inset). *N* > 5 embryos analyzed.

**Kitl-tdTomato is expressed in endothelial and mesenchymal cells in the embryonic hematopoietic microenvironment**

The requirement for Kitl in the normal development of YS-derived EMPs and AGM HSCs raised the question of what cell types produce Kitl in the embryonic hematopoietic niches. To assess this, we used a recently generated *Kitl*-tdTomato transgenic reporter mouse line [50], which was found to faithfully report Kitl-producing cells in both adult [50] and embryonic cells (Fig EV4A–C). Whole-mount immunofluorescence analysis on E8.5 *Kitl*-tdTomato transgenic concepti showed that Kitl was most strongly expressed in the para-aortic splanchnopleura (PAS) and the allantois, the tissues giving rise to the AGM and umbilical vessels, respectively. Kitl expression was also detected in YS blood islands (BI), albeit at an overall lower level (Fig 6A). A closer examination revealed expression in endothelial cells of the paired dorsal aortae (Fig 6B, arrowheads) and their

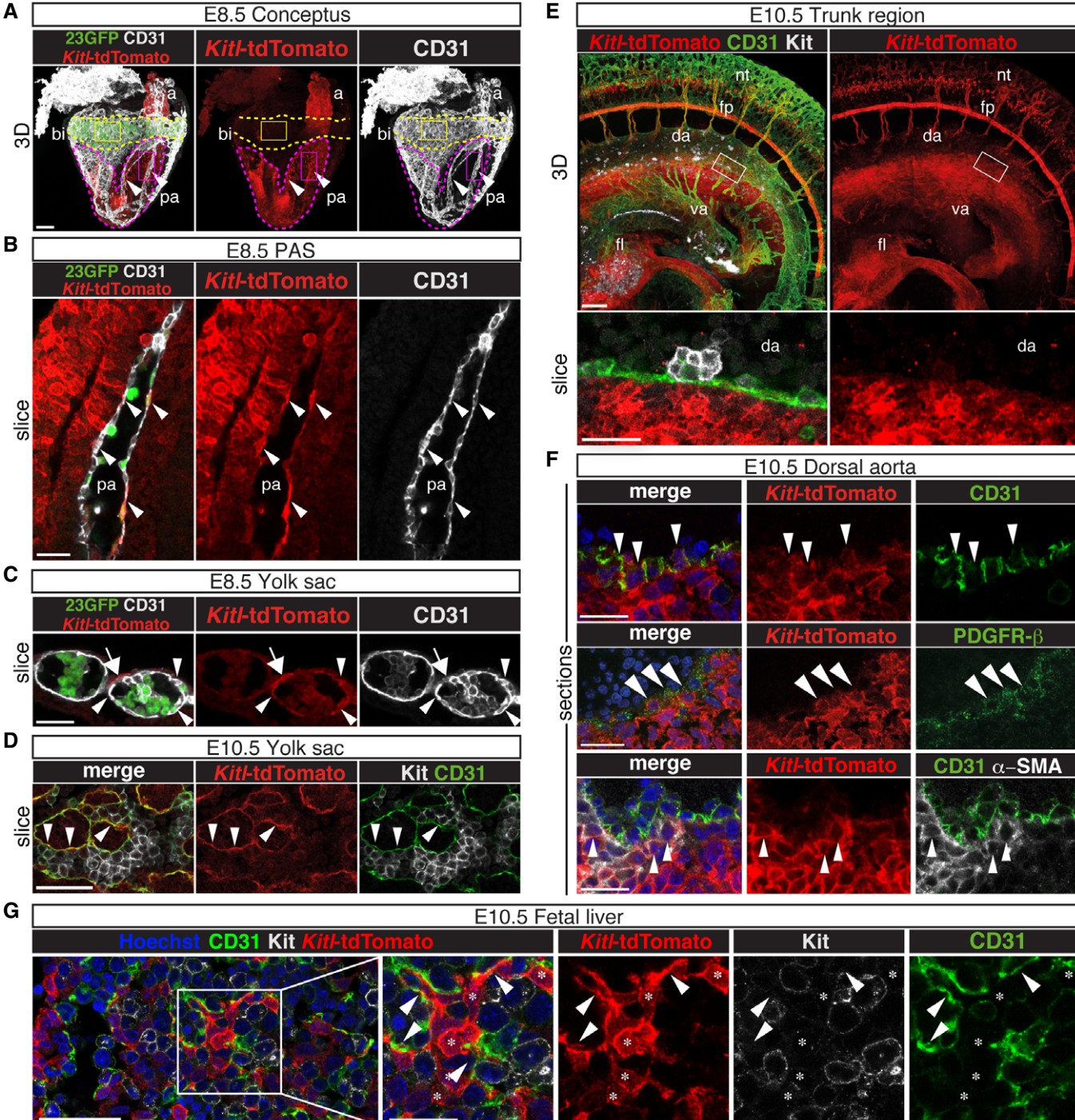

**Figure 6.**

surrounding mesenchyme. In the YS BI, Kitl was expressed in endothelial cells surrounding primitive erythroblasts (Fig 6C, arrowheads) and in some perivascular cells (Fig 6C, arrow). Endothelial expression was still evident in the E10.5 YS, around clusters of Kit$^+$ EMPs (Fig 6D, arrowheads; Fig EV4C). Approximately 20% of E10.5 YS endothelial cells (VE-Cadherin$^+$ Ter119$^-$ Kit$^-$), 15% of mesenchymal cells, and just 5–7% of Kit$^+$ cells expressed *Kitl*-tdTomato (Fig EV4D). In the E10.5 embryo, Kitl expression was apparent in the FL, and in the mesenchyme ventral to the dorsal aorta throughout the AGM area (Fig 6E). A similar Kitl expression pattern was seen in the PAS of the E9.5 embryo (Fig EV4E). *Kitl*-tdTomato was also expressed in non-hematopoietic tissues such as the floor plate and intersomitic vessels. In the E10.5 AGM, CD31$^+$ aortic endothelial cells and α-SMA/PDGFR-β$^+$ perivascular smooth muscle cells expressed *Kitl*-tdTomato (Fig 6F), with approximately 16% of AGM endothelial cells, 9% of mesenchymal cells, and few VE-Cad$^+$ Kit$^+$ hematopoietic cluster cells expressing *Kitl*-tdTomato (Fig EV4F). In the E10.5 FL, *Kitl*-tdTomato was expressed widely among mesenchymal cells/hepatoblasts and endothelium (Figs 6G and EV4G). Thus, in the embryonic hematopoietic niches, Kitl is expressed in endothelial and mesenchymal cells from the onset of YS hematopoiesis throughout the time of AGM HSC emergence and FL hematopoiesis.

### Endothelial cells are a functionally relevant source of Kitl in the YS and AGM niche

Because Kitl was widely expressed in the endothelium of YS and AGM, and endothelial cells are an important Kitl-expressing niche in the adult BM [22], we deleted Kitl in Tie2-expressing endothelium [22] of the YS and AGM niche (Fig 7A). In the YS, deletion of Kitl in Tie2-expressing endothelium resulted in a significant, approximately twofold reduction in phenotypic E11.5 EMPs (Fig 7B, left panel), functionally reflected in a twofold decrease in YS clonogenic progenitors (Fig 7C). In the E11.5 AGM, phenotypic pre-HSC II/HSCs were decreased upon deletion of Kitl from the endothelium, while pre-HSC I again remained apparently unaffected (Fig 7B, middle panel). Transplantation of Tie2-Cre::*Kitl*$^{\Delta/\Delta}$ AGM+VU cells resulted in fewer reconstituted mice than transplantation of control cells (Figs 7D and EV5A). Taken together, the hematopoietic defects in the Tie2-Cre::*Kitl*$^{\Delta/\Delta}$ YS and AGM were generally comparable to those seen in the *Sl/Sl* mutant (cf. Fig 1A and C right panels and Fig 5C,D,F).

Interestingly, in the FL, endothelial deletion of Kitl did not phenocopy the effects of the *Sl/Sl* mutation. EMPs and erythroid/clonogenic progenitors were present in normal numbers in the E11.5 Tie2-Cre::*Kitl*$^{\Delta/\Delta}$ FL (Fig 7B, right panel; Fig 7E), while these were significantly affected in the *Sl/Sl* FL (cf. Fig 2). In contrast, phenotypic Lin$^-$Sca1$^+$Kit$^+$CD48$^-$CD150$^+$ HSCs were significantly reduced in the Tie2-Cre::*Kitl*$^{\Delta/\Delta}$ E12.5 FL, similar to the *Sl/Sl* FL HSCs (Figs 7F and EV5B and C). Overall FL cellularity was not affected at E12.5, suggestive of an absence of major erythroid defects also at this time point (Fig 7G). Our data indicate a different requirement for endothelial-derived Kitl among the EMP and HSC lineages: EMPs require endothelial Kitl only as they are generated in the YS, while the HSC lineage shows a requirement from its emergence in the AGM through to early FL stages.

### Kitl/Kit interaction activates common downstream pathways in AGM and YS-derived hematopoietic cells, which are interpreted in cell type-specific ways

To begin to address whether Kitl functions by identical or distinct mechanisms in YS, AGM, or FL hematopoiesis, we examined whether the phosphoinositide 3-kinase (PI3K)/Akt pathway and/or the mitogen-activated protein kinase (MAPK)/Erk pathway were activated in E10.5 AGM hematopoietic clusters, YS EMPs, and FL hematopoietic cells. Kitl/Kit signaling is known to act through either of these pathways in a wide variety of cell types [23,51]. We chose E10.5 for our analyses as at this stage the absence of Kitl has a clear phenotype in all hematopoietic tissues, as described above. Immunofluorescence analysis of wild-type tissues showed activation of both the PI3K/Akt and MAPK/Erk pathways in hematopoietic cells at all three embryonic sites (Fig 8A and B). Interestingly, only *Sl/Sl* Kit$^+$ AGM cell clusters and YS EMPs showed an approximately twofold reduction in the level of phosphorylated Akt and Erk (Fig 8A–D), indicating that Kitl/Kit interaction is required in both these niches for activation of the PI3K and MAPK signaling cascades.

In a parallel approach, we analyzed the expression of genes known to act downstream of Kit/Kitl signaling, including genes involved in proliferation, apoptosis, differentiation, and epigenetic regulation (Appendix Table S5). Several genes were differentially expressed between cell types, though few were significantly affected by loss of Kitl (Fig 8E and Appendix Fig S3). Of these, the PI3K target *BTG1*, previously shown to be repressed by Kitl and able to exert an anti-proliferative effect in erythroid cell lines [52], was significantly upregulated in erythroid cells only (YS EMPs and FL erythroid progenitors; Fig 8E). The cell cycle inhibitor *Cdkn1b* (also known as p27kip-1), shown to be downregulated by Kitl in a myeloid cell line [53], was specifically upregulated in *Sl/Sl* YS EMPs. *Cdkn1a* (p21) displayed a similar trend (Fig 8E). FL erythroid precursors showed downregulation of *Stat5a*, known to promote erythroid cell proliferation and differentiation downstream of Kit [54], and *Myc*, which was described to mediate cell cycle progression in erythroblasts [55] and was recently shown to be downregulated in Kit W$^{41}$ mutants underlying their proliferative defects [56] (Fig 8E). In addition, we observed an upregulation in the anti-apoptotic gene *Bcl2l1* (Bcl-xL) in FL populations (Appendix Fig S3). *Bax* and *Bcl2* expression was significantly affected in (pre-)HSCs as described above (Fig 5J), while we did not observe consistent changes in expression of the various cell cycle-associated genes in these cells: only expression of *Ccnd1*, but none of the others (*Myc*, *Cdkn1a*, *Cdkn1b*, *Ccnd2*) was significantly affected in pre-HSC II/HSC (Fig 8E, Appendix Fig S3). Taken together, these data suggest that the common PI3K/Akt and MAPK/Erk pathways that act downstream of Kitl-Kit signaling in YS and AGM are interpreted differently in (pre-)HSC and erythroid cells. The specific gene interactive networks underlying the effects of Kit in each cell type remains to be established.

## Discussion

Few studies to date have examined embryonic hematopoietic niches *in vivo*, hampering direct comparison between adult and embryonic niche composition [20,21]. Here, we examined the requirement for

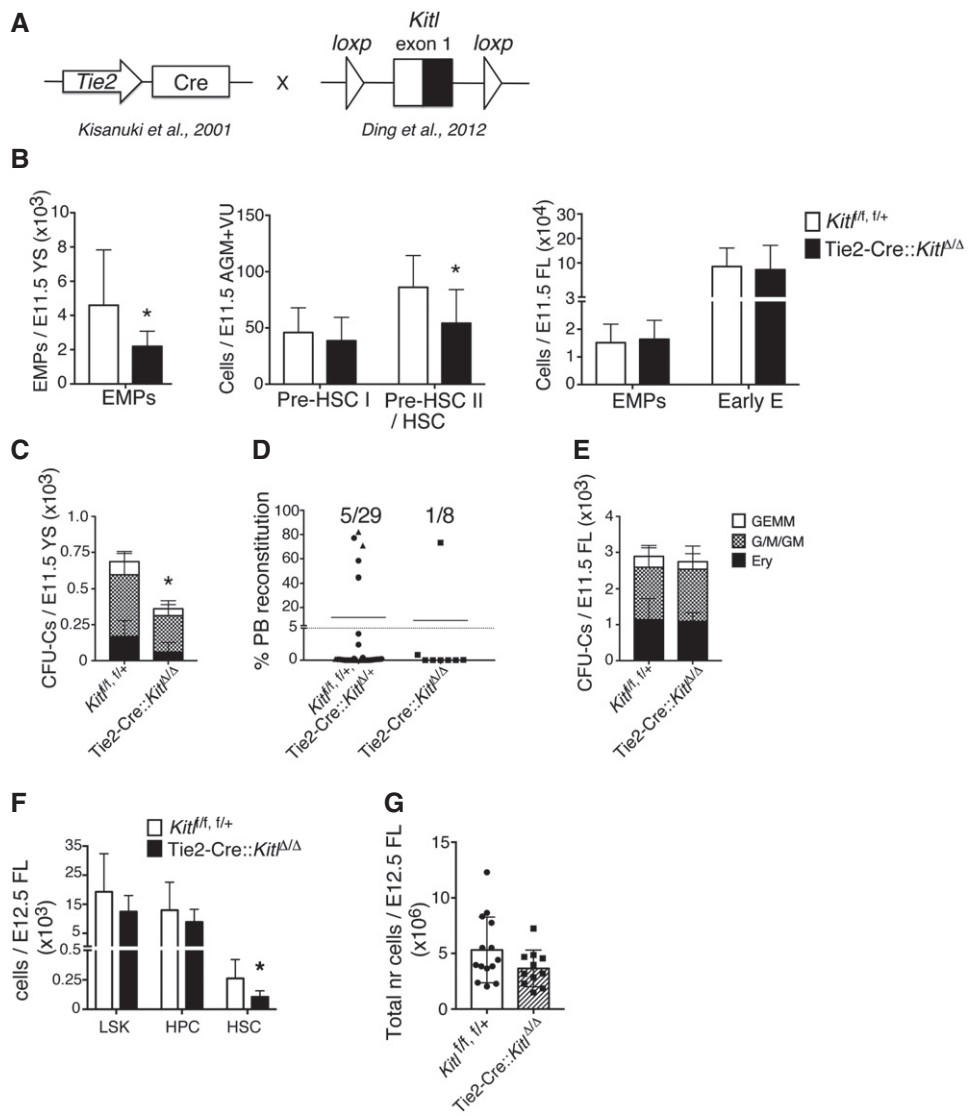

**Figure 7. Endothelial Kitl is required for hematopoietic stem and progenitor cells in the AGM and for EMPs in the YS.**

A  Schematic showing the two genetically modified mouse lines crossed to perform conditional deletion of Kitl in endothelial cells.

B  Left: number of Ter119⁻ Kit⁺ CD41⁺ CD16/32⁺ EMPs in Tie2cre::Kitl^Δ/Δ and control Kitl^f/f or f/+ E11.5 YS, determined by flow cytometry. Data are the mean (± SD) of eight control and seven Tie2cre::Kitl^Δ/Δ YS analyzed individually over four independent experiments. Tail somite range: 13–16. Middle: number of pre-HSC type I (Ter119⁻ VE-Cadherin⁺ Kit⁺ CD41⁺ CD45⁻) and pre-HSC type II/HSC (Ter119⁻ VE-Cadherin⁺ Kit⁺ CD41⁺ CD45⁺) in Tie2cre::Kitl^Δ/Δ and control Kitl^f/f or f/+ E11.5 AGM+VU, determined by flow cytometry. Data are the mean (± SD) of seven control and five Tie2cre::Kitl^Δ/Δ AGM+VU analyzed individually over three independent experiments. Tail somite range: 14–15 (Kitl^f/f, f/+); 14–16 (Tie2cre::Kitl^Δ/Δ). Right: number of Ter119⁻ Kit⁺ CD41⁺ CD16/32⁺ EMPs and Ter119⁻ Kit⁺ CD41⁻ CD71⁺ CD44⁺ early erythroid cells in Tie2cre::Kitl^Δ/Δ and control Kitl^f/f or f/+ E11.5 FL, determined by flow cytometry. Data are the mean (± SD) of seven control and five Tie2cre::Kitl^Δ/Δ FL analyzed individually over three independent experiments. Tail somite range: 13–16.

C  CFU-C numbers in Tie2cre::Kitl^Δ/Δ and control Kitl^f/f or f/+ E11.5 YS. Data are the mean (± SD) of six control and three Tie2cre::Kitl^Δ/Δ YS plated in duplicate in two independent experiments. Tail somite range: 13–16.

D  Analysis of long-term multi-lineage HSC potential in E11.5 Tie2cre::Kitl^Δ/Δ and control Tie2cre::Kitl^Δ/+ or Kitl^f/f or f/+ AGM+VU. Irradiated CD45.1 syngeneic mice were transplanted with 1 e.e. of AGM+VU cells. PB chimerism is represented as the percentage of donor CD45.2⁺ cells among total CD45⁺ cells, 16 weeks after transplant. A total of 29 recipients were transplanted with Tie2cre::Kitl^Δ/+ or Kitl^f/f or f/+ cells and eight with Tie2cre::Kitl^Δ/Δ cells, over seven independent experiments. Kitl^f/f or f/+ is represented with a circle, Tie2cre::Kitl^Δ/+ a triangle and Tie2cre::Kitl^Δ/Δ with a square. Tail somite range: 12–17 (control); 12–17 (Tie2cre::Kitl^Δ/Δ).

E  CFU-C numbers in Tie2cre::Kitl^Δ/Δ and control Kitl^f/f or f/+ E11.5 Fl. Data are the mean (± SD) of 7 control and 5 Tie2cre::Kitl^Δ/Δ FL plated in duplicate in three independent experiments. GEMM: granulocyte, erythroid, monocyte/macrophage, megakaryocyte; G/M/GM: granulocyte, monocyte/macrophage; Ery: erythroid. Tail somite range: 13–16.

F  Number of phenotypic LSK, HPC, and HSC per FL in E12.5 Tie2cre::Kitl^Δ/Δ and control (Kitl^f/f or f/+) FL, determined by flow cytometry and total FL cell counts. Cells were gated as 7-AAD⁻ Lin⁻ (F4/80⁻ CD3e⁻ Nk1.1⁻ Ter119⁻ Gr-1⁻ B220⁻ CD19⁻). LSK: Lin⁻ Sca1⁺ Kit⁺; HPC: Lin⁻ Sca1⁺ Kit⁺ CD48⁺ CD150⁻ and HSC: Lin⁻ Sca1⁺ Kit⁺ CD48⁻ CD150⁺. Gating strategy is shown in Fig EV4E. Data are the mean (± SD) of 12 control and 6 Tie2cre::Kitl^Δ/Δ FLs analyzed individually over three independent experiments.

G  Total cellularity of E12.5 Kitl^f/f or f/+ (n = 14) and Tie2cre::Kitl^Δ/Δ (n = 11) FL. Error bars represent SD.

Data information: *P < 0.05 (unpaired two-tailed Student's t-test).

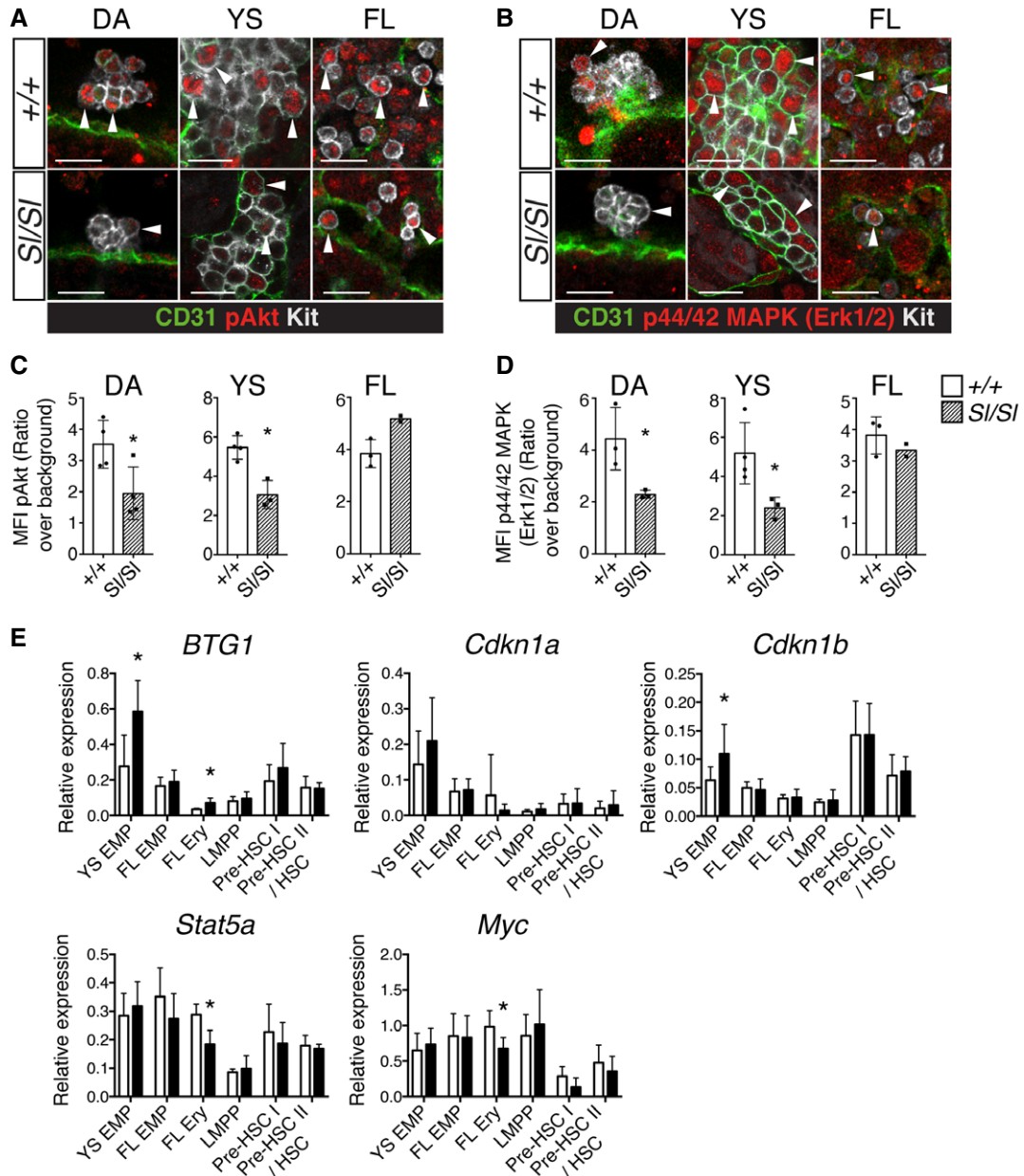

**Figure 8. Kitl is required for Akt and Erk phosphorylation in the AGM and YS.**

A, B Confocal whole-mount immunofluorescence analysis of phosphorylated Akt (pAkt) (A) and phosphorylated p44/42 MAPK (pErk1/2) (B) in Kit$^+$ cells of wild type (32–36 sp) and *Sl/Sl* (31–35 sp) E10.5 dorsal aorta, yolk sac, and fetal liver. Images are 2.5-µm-thick single longitudinal *Z*-slices; dorsal up (DA). Arrowheads point to Kit$^+$ cells with varying levels of pAkt and pErk1/2 expression. A total of two wild type and two *Sl/Sl* embryos for each staining were analyzed. Scale bars: 20 µm.

C, D Quantification of the immunofluorescence in (A, B), showing mean fluorescence intensity (MFI) ratio of pAkt and p44/42 MAPK (pErk1/2) within wild type (+/+) and *Sl/Sl* CD31$^+$ Kit$^+$ dorsal aorta (DA), yolk sac (YS), and fetal liver (FL) hematopoietic cells. MFI values were normalized to background in the adjacent ventral mesenchyme for DA, to erythrocytes in the YS or stroma in the FL. Additional quantification was performed on sections from two wild type and two *Sl/Sl* embryos of similar age range. MFI measurements were taken using ImageJ. Number of cells measured for pAkt: 100 hematopoietic cells and 73 background (+/+); 79 hematopoietic and 52 background (*Sl/Sl*) in the AGM; 127 hematopoietic cells and 95 background (+/+); 110 hematopoietic and 82 background (*Sl/Sl*) in the YS; 102 hematopoietic cells and 78 background (+/+); 78 hematopoietic and 83 background (*Sl/Sl*) in the FL. Number of cells measured for pErk1/2: 94 hematopoietic cells and 82 background (+/+); 114 hematopoietic and 91 background (*Sl/Sl*) in the AGM; 174 hematopoietic cells and 109 background (+/+); 123 hematopoietic and 82 background (*Sl/Sl*) in the YS; 100 hematopoietic cells and 70 background (+/+); 63 hematopoietic and 57 background (*Sl/Sl*) in the FL. *N* = 4 (+/+), 4 (*Sl/Sl*) embryos analyzed for DA and YS; *N* = 3 (+/+), 2 (*Sl/Sl*) for FL. Error bars represent SD.

E Multiplex Fluidigm qRT–PCR analysis of pre-HSC type I (Ter119$^-$ VE-Cadherin$^+$ Kit$^+$ CD41$^+$ CD45$^-$), pre-HSC type II/HSC (Ter119$^-$ VE-Cadherin$^+$ Kit$^+$ CD41$^+$ CD45$^+$), YS and FL EMP (Ter119$^-$ Kit$^+$ CD41$^+$ CD16/32$^+$), FL Early E (Ter119$^-$ Kit$^+$ CD41$^-$) and peripheral blood LMPP (Lin$^-$ CD19$^-$ B220$^-$ CD45$^+$ Kit$^+$ Flt3$^+$ IL7Rα$^+$), isolated from E11.5 (11–17 tail somites) wild type and *Sl/Sl* embryos. Four biological replicates for wild type (9 total embryos) and five biological replicates (9 total embryos) for *Sl/Sl* were analyzed. Data are mean (± SD).

Data information: *P < 0.05 (unpaired two-tailed Student's *t*-test).

Kitl, an important hematopoietic regulator in adult BM, in the in vivo YS and AGM hematopoietic niche. Analysis of Kitl-deficient (Sl/Sl) mouse embryos revealed a previously unrecognized requirement for Kitl in YS EMPs, manifested in reduced EMP expansion upon their emergence in the YS. EMPs continue to require Kitl in the FL, where the loss of Kitl resulted in severe and early defects in EMP-derived erythropoiesis. Indeed, our data indicate that the anemia of Sl/Sl embryos is, at least initially, attributable to a defect of YS-derived EMPs. This is in apparent contrast to earlier studies that attributed the anemia to a defect in HSC-derived progenitors, and might be because previously YS and early FL hematopoiesis were not comprehensively analyzed [32,35,36]. We found that in addition to EMPs, YS-derived LMPPs also require Kitl for their normal development.

Interestingly, Kitl-deficient embryos showed a marked reduction in macrophages in the E10.5 AGM and E14.5 skin, lungs, limb buds, and fetal liver, but not in the brain. As fetal macrophages do not express Kit receptor [47,57], and EMPs were previously shown to contribute to tissue-resident macrophage populations [43], the fetal macrophage depletion in Sl/Sl embryos is most likely a consequence of the EMP expansion defect in the YS and/or FL. Importantly, it has been disputed whether EMPs contribute to tissue-resident macrophages in the brain (known as microglia), and Kitl-independent primitive macrophages have alternatively been proposed as their cells of origin [47,58,59]. As in our study macrophages in the brain were unaffected, our data support the view that microglia have a distinct origin from other fetal tissue macrophages that primarily originate from EMPs and/or fetal HSCs that rely on Kitl [43,47,48].

We found that macrophages scattered throughout the AGM were decreased in the Sl/Sl embryo. As they have been implied as a niche cell type [60], their reduction could potentially affect AGM hematopoiesis. However, the macrophages that are closely associated with the hematopoietic cell clusters of the dorsal aorta were present in normal numbers at the time of HSC emergence, making it unlikely they are indirectly responsible for the observed Sl/Sl AGM phenotype.

In the in vivo AGM region, Kitl affected the maturation and survival of the HSC lineage, after its divergence from hemogenic endothelium. Specifically, and in contrast to in vitro studies [11,26], our data revealed that Kitl was predominantly required at the level of the pre-HSC II/HSC and not the pre-HSC I. Niche factors other than Kitl may partly compensate for the loss of Kitl/Kit signaling in pre-HSC I. Candidates include thrombopoietin (TPO) and Oncostatin M, a member of the IL-6 family of cytokines. TPO/Mpl signaling is known to affect the generation/expansion of HSCs in the AGM [61] and acts synergistically with Kitl in mouse BM HSPCs [62]. Oncostatin M has been reported to promote proliferation of AGM-derived HSPCs in vitro [63], and in zebrafish was recently shown to synergize with kitlgb (one of the two zebrafish paralogs of Kitl) to promote the expansion of HSCs in the caudal hematopoietic tissue (CHT), the zebrafish FL equivalent [64]. The precise action of these factors on mouse AGM pre-HSCs/HSC and their interaction with Kitl awaits further experimentation.

Within the YS, AGM, and FL niches, Kitl was expressed in different stromal compartments, including endothelial cells in the vicinity of HSPCs. Deletion of Kitl in Tie2-positive endothelial cells was sufficient to phenocopy the Sl/Sl hematopoietic defects of the YS and AGM. In contrast, in the FL, endothelial deletion of Kitl only recapitulated the defect in the phenotypically defined HSC lineage,

and no defect was observed in FL EMPs/erythroid progenitors. The latter implies that FL EMPs obtain Kitl from a different niche cell type, as they are severely affected in the Sl/Sl embryo. A strong candidate is the hepatoblast that we found to be the most prevalent source of Kitl in the FL, consistent with previous observations based on gene expression [65,66]. Altogether, our data demonstrate changing requirements for endothelial Kitl in the sequential EMP niches, but not HSC niches, during embryonic development.

The critical role for endothelial cells in providing Kitl for developing EMPs in the YS and for (pre-)HSCs in the AGM and FL niches is in line with endothelial cells being an important source of Kitl for mouse BM HSCs [22] and HSCs in the zebrafish CHT [67], and suggests a conserved role for endothelial cells in delivering Kitl to HSCs, regardless of their developmental stage. Kitl occurs in either a soluble or a membrane-bound form, generated by alternative splicing of a precursor RNA [68]. Both can activate the Kit receptor, although the membrane-bound form was reported to yield a stronger signal through increased tyrosine kinase activation [69]. Membrane-bound, but not soluble Kitl, induces long-term proliferation of $CD34^+$ cells [70] and survival of early thymic progenitors [50]. Thus, it is plausible that the dependency of YS EMPs and AGM HSCs on endothelial Kitl reflects a requirement for membrane-bound Kitl by these cells. Alternatively, soluble Kitl may act in a local niche manner. Conditional deletion of membrane-bound Kitl will help distinguish between these possibilities.

Our data suggest that the mechanisms downstream of Kitl/Kit signaling in different embryonic hematopoietic populations are complex and context-dependent. While activation of the PI3K and MAPK pathways in YS EMPs and the AGM HSC lineage is dependent on Kitl, the functional consequences of Kitl/Kit signaling in these cells were different. Our data indicate that in YS EMPs Kitl/Kit signaling normally represses the expression of anti-proliferative genes such as BTG1 and Cdkn1b, as these genes were upregulated in the Sl/Sl cells. This is in line with the reduced proliferation of these cells. In contrast, we did not detect widespread differences in cell cycle of pre-HSCs/HSC in wild type versus Sl/Sl embryos, either by BrdU or gene expression analysis. Conversely, Kitl appeared to be required in pre-HSCs/HSCs for promoting their survival, in accordance with its role in the adult BM [22,40]. This effect could be mediated by PI3K/Akt, as this signaling cascade was shown to promote cell survival [71], also in response to Kit activation [72]. Furthermore, in the absence of Kitl we observed upregulation of the pro-apoptotic gene Bax and downregulation of the anti-apoptotic gene Bcl2 in the HSC lineage. Of interest, using in vitro culture systems followed by functional assays, expansion of the emerging HSC lineage was recently reported to occur through proliferation of pre-HSCs, rather than continued de novo generation of these cells [45,73]. Our data indicate that Kitl does not play a significant role in regulation of proliferation during the generation of the HSC lineage, but affects pre-HSC/HSC numbers through protection from apoptosis.

In conclusion, our study uncovered a critical role for endothelial Kitl in the normal expansion of YS-derived EMPs, and showed its requirement in the survival of the AGM HSC lineage in vivo. In addition, we provide evidence for an early defect in FL erythropoiesis downstream of the EMP defect and showed that fetal tissue macrophages (apart from microglia) were affected. It will be of interest to explore the gene interactive networks downstream of Kitl in the HSC lineage to further explore inroads into manipulating the

generation of these cells from pluripotent stem cells. Of equal relevance, exploring what other microenvironmental factors are produced by Kitl-expressing endothelial niche cells is expected to reveal new extrinsic regulators of HSC generation.

# Materials and Methods

### Mice

All procedures involving mice were in compliance with UK Home Office regulations and approved by the Oxford University Clinical Medicine Ethical Review Committee or by the University of Rochester Committee on Animal Resources. Steel [37–39], Kitl-tdTomato [50], 23GFP [2,74], Kitl^flox [22], and Tie2-Cre [75] mice were housed with free access to food and water and maintained in a 12-h light–dark cycle. 23GFP mice were maintained on a mixed (CBAxC57BL/6) background, others on a CD45.2 C57BL/6 genetic background. Genotyping primers are listed in Appendix Table S1. Timed pregnancies were generated, and embryos collected and dissected, as previously described [2]. E8.5 to E10.5 embryos were staged according to somite pairs, E11.5 embryos based on tail somite pairs [76], and older embryos on the basis of morphological criteria (E12.5–E14.5).

### Embryonic blood collection

Concepti were collected in PBS supplemented with 10% FBS and PS (37°C). Umbilical vessels were clamped while removing the placenta and embryos transferred to individual dishes to bleed out. Medium was collected and cells spun down for counting and analysis.

### Cytospin preparation and Wright-Giemsa's staining

Cytospins were prepared on a Shandon Cytospin 4 (Thermo Scientific) and were stained with a Wright-Giemsa's stain using an automated HEMATEK slide-staining machine (Bayer Healthcare). Images were acquired on a Nikon Eclipse e600 microscope equipped with a Nikon DXM1200C camera, using a PlanFluor 40×/0.75 objective.

### Immunofluorescence analysis and imaging

Whole-mount immunofluorescence on embryos and yolk sac was performed as previously described [77]. Images were acquired on a Zeiss AXIO Examiner.Z1 upright microscope equipped with a Zeiss LSM-780 confocal system, using a 25× NA:0.8 DIC Imm Kor UV VIS-IR objective or on a Zeiss AXIO Observer.Z1 inverted microscope equipped with a Zeiss LSM-880 confocal system using an 25× LDLCI PlnApo NA:0.8 DI or a 40× C Apo 1.1W DICIII objective. Imaging was performed at room temperature. Images were processed using IMARIS v7.5.0 (Bitplane), Zeiss Zen and Adobe Photoshop CS6. 3D reconstructions are maximum intensity projections. Appendix Table S2 lists antibodies used for immunofluorescence analysis.

### Flow cytometry and cell sorting

Single cell suspensions were generated and processed for flow cytometry as previously described [2]. Analysis was carried out on BD LSR II, BD LSR Fortessa or BD LSR Fortessa X-20 analyzers. Cell sorting was performed on a BD FACSAria Fusion sorter using a 100 μm nozzle. Compensation and gates were set using unstained, single stained and fluorescence-minus-one (FMO) controls. Dead cells were excluded based on Hoechst 33258 (Sigma) or 7-AAD (Sigma) incorporation. For imaging flow cytometry, FL cells were stained and labeled as previously described [78], run on a ImageStreamX (Millipore Sigma), and data analyzed with IDEAS software (Millipore Sigma) following the gating strategy previously described [79]. Appendix Table S3 lists antibodies, conjugates, and DNA dyes used for flow cytometry.

### Single and multiplex Quantitative Real-Time PCR (qRT–PCR)

For single gene qRT–PCR, total RNA was isolated from sorted cells using the Quick-RNA Microprep Kit (Zymo Research), according to manufacturer's instructions. RNA was quantified on a Nanodrop ND-1000 spectrophotometer (Thermo Scientific). Reverse transcription was performed using Superscript III First-Strand Kit (Life Technologies). qRT–PCR was performed using TaqMan probes (Applied Biosystems) on an ABI PRISM 7000 system (Applied Biosystems). −RT samples were used as controls. Data were analyzed using the ΔCt method and are shown as $2^{-\Delta Ct}$.

Multiplex qRT–PCR was performed as previously described [2]. Pools of 10 cells were sorted directly into −RT/preamplification mix and analyzed on a Biomark platform (Fluidigm). Data were normalized to Atp5a1, Hprt, and Ubc. A list of Taqman probes used for qRT–PCR is shown in Appendix Table S4. Appendix Table S5 lists genes used in multiplex qRT–PCR and references to previous work in relation with Kitl.

### Cell cycle analysis

2 mg of a 10 mg/ml BrdU solution was injected intra-peritoneally into pregnant females. Two hours later, embryos were collected and BrdU incorporation analyzed by FITC BrdU Flow Kit (BD Pharmingen), following manufacturer's instructions.

### CFU-C (Colony-forming unit-culture) assays

CFU-C assays were performed using Methocult M3434 (mixed colonies) or M3334 (CFU-E) (Stem Cell Technologies). Cells were plated in duplicate and cultured at 37°C, 5% $CO_2$. Colonies were scored after 2 (M3334) or 7 days (M3434).

### Repopulation assays

Single-cell suspensions of CD45.2$^+$ AGM+VU, along with 200,000 CD45.1$^+$ spleen carrier cells, were injected into CD45.1$^+$ 8-week-old adult male 9Gy-conditioned recipients (split dose; $^{137}$Cs). Donor-derived chimerism was determined by flow cytometry in peripheral blood (PB) at 6 and 16 weeks post-transplantation. PB, treated with BD Pharmlyse (BD), was labeled with anti-CD16/32 (Fc block), anti-CD45.1-APC, and anti-CD45.2-FITC. Long-term multi-lineage reconstitution levels in PB, BM, spleen, and thymus were determined by flow cytometry at 16 weeks; antibodies are listed in Appendix Table S3. Recipients showing ≥ 5% donor-derived cells were considered reconstituted. Repopulation levels in reconstituted mice were compared using Mann–Whitney U-test.

## RNA sequencing

Samples of 100 cells each were processed using SMARTSEQ2 [80]. Paired end sequencing (75 bp) was carried out on an Illumina HiSeq 4000. Reads were aligned to the NCBI37/mm9 mouse reference genome using STAR (v2.4.2a). Lane replicates for each sample were merged into one BAM file. Read counts for each gene were determined by using the featureCounts program from the Subread software package (v1.4.5) and processed/normalized using the DESeq2 package (v1.12.4). *P*-values were corrected for multiple testing using the Benjamini and Hochberg method, and differentially expressed genes were identified as those with false discovery rate (FDR) below 0.1. Heatmaps were generated in R, and GO analysis was performed using MetaCore (Thomson Reuters).

## Statistics

Unpaired two-tailed Student's *t*-test assuming equal variance was used to determine the level of significance (unless otherwise indicated). $P < 0.05$ was considered statistically significant (indicated by an asterisk).

# Data availability

RNA-Seq data are available in the GEO database with the accession number GSE105267.

**Expanded View** for this article is available online.

## Acknowledgements

The authors would like to thank current and past members of the de Bruijn laboratory for practical advice, and in particular Christina Rode, Stella Antoniou, Lucas Greder; Tiago C. Luis and Mario Buono for advice; Christoffer Lagerholm at the Wolfson Imaging Centre Oxford for technical expertise and assistance with imaging; Kevin Clark and Paul Sopp at the WIMM FACS facility for providing cell sorting services and technical expertise; Emmanouela Repapi at the Computational Biology Research Group (CBRG); the Biomedical Services staff for animal care. This work was supported in Oxford by a programme in the MRC Molecular Hematology Unit Core award (Grant number: MC_UU_12009/2); JP was supported in Rochester by NIH funds (1R01 HL130670). The Wolfson Imaging Centre Oxford is supported by the Medical Research Council via the WIMM Strategic Alliance (G0902418), the Molecular Haematology Unit (MC_UU_12009), the Human Immunology Unit (MC_UU_12010), the Wolfson Foundation (Grant 18272), and by an MRC/BBSRC/EPSRC grant (MR/K015777X/1) to MICA—Nanoscopy Oxford (NanO): Novel Super-resolution Imaging Applied to Biomedical Sciences, Micron (107457/Z/15Z). The facility was supported by WIMM Strategic Alliance awards G0902418 and MC_UU_12025. The WIMM Flow Cytometry facility is supported by the MRC HIU; MRC MHU (MC_UU_12009); NIHR Oxford BRC and John Fell Fund (131/030 and 101/517), the EPA fund (CF182 and CF170) and by the WIMM Strategic Alliance awards G0902418 and MC_UU_12025. We thank Neil Ashley for his help with library preparation. The WIMM Single Cell Core Facility was supported by the MRC MHU (MC_UU_12009), the Oxford Single Cell Biology Consortium (MR/M00919X/1), and the WT-ISSF (097813/Z/11/B#) funding. The facility was supported by WIMM Strategic Alliance awards G0902418 and MC_UU_12025.

## Author contributions

EA designed and led the study, performed most experiments, analyzed and interpreted data, made figures, wrote the manuscript; VF performed flow cytometry and transplantation experiments and analyzed data; KEM performed imaging flow cytometry, analyzed and interpreted data, and edited the manuscript; JH assisted with experiments and analyzed RNA-Seq data; JC performed some flow cytometry analysis; CN contributed the *Kitl*-tdTomato mice and interpreted data; JP interpreted data and edited the manuscript; SEWJ interpreted data and edited the manuscript; MFTRB led and supervised all aspects of the study, analyzed and interpreted data, and wrote the manuscript.

## Conflict of interest

The authors declare that they have no conflict of interest.

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
