## [Review Process File · EMBO Reports]

Kit ligand has a critical role in mouse yolk sac and aorta-gonad-mesonephros hematopoiesis

Emanuele Azzoni, Vincent Frontera, Kathleen E. McGrath, Joe Harman, Joana Carrelha, Claus Nerlov, James Palis, Sten Eirik W. Jacobsen, Marella F.T.R. de Bruijn

Review timeline:	Submission date:	10th Nov 17
	Editorial Decision:	19th Jan
	Revision received:	4th Jul 18
	Editorial Decision:	20th Jul
	Revision received:	24th Jul 18
	Accepted:	27th Jul 18

Editor:

Transaction Report:

1st Editorial Decision

19th Jan

Thank you for the submission of your research manuscript to EMBO reports. We have now received reports from the three referees that were asked to evaluate your study, which can be found at the end of this email.

As you will see, all three referees state that the work was carefully executed, however they also have raised a number of concerns and suggestions to improve the manuscript, or to strengthen the data and the conclusions drawn. As the reports are below, I will not detail them here. However, it will be important to revise the manuscript and to clearly focus on the new aspects of the work, as all three referees pointed out that the conceptual advance of the study seems rather limited. In order to provide further insight, we think that the additional comments of referee #1, and the three major points of referee #2 also need to be addressed, best with additional data.

Given the constructive referee comments, we would like to invite you to revise your manuscript with the understanding that all referee concerns must be addressed in the revised manuscript and in a point-by-point response. Acceptance of your manuscript will depend on a positive outcome of a second round of review. It is EMBO reports policy to allow a single round of revision only and acceptance or rejection of the manuscript will therefore depend on the completeness of your responses included in the next, final version of the manuscript.

REFeree COMMENTS

Referee #1:

The manuscript by Azzoni et al. addresses the role of Stem Cell Factor (SCF), the established ligand of the cKit receptor tyrosine kinase, in hematopoiesis in the yolk sac and AGM. The authors cite data from the 1990s to make an argument that it is unclear whether SCF has important functions pre-fetal liver hematopoiesis. They establish that SCF is important, and endothelial cells constitute an important source of SCF. The authors conclude that SCF is a "critical regulator of HSPC development in the AGM and YS endothelial niche". HSPCs express cKit in all contexts studied - in vivo and in vitro. In an extremely well-established paradigm, SCF activates cKit to induce signaling that controls HSPC generation and function. The finding the SCF also has important functions pre-fetal liver hematopoiesis is not surprising, given the cKit⁺ hematopoietic precursors in this context. While it is useful to know that endothelial cells represent an important source of SCF at this stage, endothelial cells were known to secrete SCF. In summary, the current work appears to have been carefully executed, but the results only incrementally advance existing knowledge in the field. Furthermore, it is not evident that broader insights emerge from the studies.

Additional Comments:

- 1) Evidence was provided that SCF "has a previously unrecognized role in YS EMP proliferation". This conclusion is based on a ~2 fold change in proliferation. Does this involve a unique mechanism, or does it recapitulate known mechanisms by which SCF/cKit control proliferation? In addition, the magnitude of the change seems quite small. Is that because SCF is only a minor proliferative determinant?
- 2) SCF depletion impaired EMPs, tissue macrophages, LMPPs and HSCs. Does SCF function via identical mechanisms in these contexts, or are common SCF-derived signals interpreted differently? Providing new mechanistic insights would have potential to yield important results - results that might broadly inform SCF-cKit mechanisms.
- 3) In the yolk sac, is Tie2-CRE only active in endothelial cells? Do endothelial cells generate the majority of the SCF in the yolk sac, or is endothelial cell-derived SCF uniquely important?

Referee #2:

This work investigates the importance of Kit-ligand (Kitl) in early embryonic hematopoiesis. The authors found that alteration of Kitl strongly reduces the population of erythroid-myeloid-progenitors originating in the YS and of the pre-HSCs in the AGM. They used three transgenic mouse lines to identify the specific expression of Kitl in the embryonic tissues and study alteration of the Kitl-dependent pathway using broad and endothelial-specific ablation.

General Comments:

The main question of the authors is of value for the field, as the factors controlling early embryonic hematopoiesis are largely unknown. Importantly, they base their analysis largely on in-vivo systems and genetic mutants. On the other hand, previous reports already defined the importance of Kitl in the AGM, in vitro (Rybtsov et al. 2014 and other previous publications) and more recently in vivo (Souilho et al 2016). This partially hampers the level of novelty of the findings here described, but the authors come to slightly different conclusions and in addition focus on the YS EMPs and primitive macrophages, which has been less intensively studied.

The paper will gain by important revisions as described in more details below.

- 1) The authors found that alteration of Kitl results in decreased number of macrophages in the majority of tissues and decreased number of cKit⁺ cells in the hematopoietic clusters of the AGM. It is possible that the two phenotypes are linked to each other, as the macrophages are present in the surrounding of the hematopoietic clusters in the AGM (Travnickova et al, 2015) and might contribute to the maintenance of their structure and functionality. Can the authors test whether macrophages are equally present in clusters in the presence or absence of kitl?
- 2) The phenotype of the SI/SI mice in the AGM reflects in decreased number of cKit⁺ cells and

decreased reconstitution potential of the AGM-derived HSCs. Have the authors counted the number of clusters? In the absence of *kitl*, are *Kit*⁺ cells reduced because clusters are smaller or there are less clusters or both? Is there more EHT, so more clusters of more individual *kit*⁺ cells in the endothelium?

3) Is the lack of *Kitl* directly blocking the proliferation of functional HSCs or is it disturbing the maturation of the hematopoietic clusters as published by the Medvinsky lab? This may be difficult to investigate, but the authors should speculate on that.

Minor points:

Figure1: Whole mounts with *kitl*-tomato are not clear. The authors choose to show the maximal projection of the embryo, but maybe a single stack image would be clearer. In this sense: In Fig 1A, is not clear to me what they want to show. Can the authors draw a line showing the conceptus? Is the YS mainly negative for *kitl*-Tomato? Figure 1C shows that there is not. Fig 1B: Whole mount imaging of the *Kitl*-Tomato transgene should be improved or supported by additional magnification of the aortic region or showing a specific stack.

Fig1D.

The claim that perivascular cells are expressing *kitl* is not well supported by the staining of SMA because at 10.5 the perivascular structures are not well defined and SMA is expressed mainly by endothelial cells. In fact the authors will conclude that *kitl* from endothelial cells is responsible for the hematopoietic effects, which makes this part more confusing. The authors should consider restructuring or remaking figure 1 for clarity.

Sup Fig1

SF1A: No clear positive populations for *kitl*-tomato are shown in the dotplots for E8.5, AGM+VU or YS. Could the authors do double staining with *kitl* antibody and check the correlation of expression? That would be really informative.

SF1B: The label on the Y-axis is unclear.

SF1D and E: The authors exclude MC with CD41 and CD45 staining. Can they show the dotplots? Are these markers used in the definition of HC or HP?

Figure 2

Fig2F: The FACS plot does not show any cell in the M phase in the dot plots, although several cells express the mitotic marker PHH3 in Fig2E. Can they reconcile both findings?

Figure6

Fig6B: Quantification lacking for apoptosis

Figure

Fig7A,B: Are CD45⁺ cells still present in the cluster after *Kitl* ablation?

Referee #3:

EMBO 45477-T "A critical role for Kit ligand in mouse pre-liver hematopoiesis" by Report-2017-de Azzoni and colleagues. The authors studied expression and function of Kit ligand (*Kitl*) in embryonic and fetal hematopoiesis. They argue initially that little is known about extrinsic factors regulating the hematopoietic system, yet in this paper the author then go on and study one of the best known and most intensively studied hematopoietic growth factors, *Kitl*. Along these lines, the argument that factors may help in supporting hematopoietic stem cell derivation or expansion *in vitro* is perhaps ok but from all we know it is clear that *Kitl* is not a factor capable of doing this. Also on the critical side, *Kit* expression has long been studied in yolk sac and subsequent stages (see for example the detailed report by Matsui, Zsebo & Hogan. Embryonic expression of a hematopoietic growth factor encoded by the *Sl* locus and the ligand for c-kit. Nature 1990).

On the other hand, one could argue that the authors have re-addressed expression and function of

Kitl in embryonic and fetal hematopoiesis, and today, i.e. 25 years later, more sophisticated tools are available and have been utilized by the authors here (e.g. genetic reporter). The data include visualization of Kitl expression in a fluorescent reporter, careful and quantitative re-analysis of hematopoiesis in natural SI (Steel; lacking both membrane bound and soluble Kitl) mutant embryos and fetuses, and endothelial cell-restricted conditional deletion of Kitl. These experiments are well done and the data seem solid.

In summary, the paper is written to suggest that very little was known about Kitl expression and function. This is exaggerated. While much of the data look like details in our overall picture in the Kit-Kitl system, the data still seem worth publishing, for as long as a revised paper focuses on the specific and new aspects shown here (and avoids construed justifications for why all of this was studied).

Minor: Formally speaking, the current nomenclature is Kit not c-Kit.

1st Revision - authors' response

4th Jul 18

We would like to thank the Editor for giving us the opportunity to revise our manuscript, which we think has been significantly improved in the revised version enclosed here. As detailed below in the point-by-point reply to each referee's comments, wherever it was possible we have addressed experimentally each of the concerns raised by the referees. Specifically, we have introduced a new set of experiments aimed at strengthening the mechanistic aspects of our work. We have assessed the activation of pathways downstream of Kit/Kitl signaling in embryonic hematopoietic cells by immunofluorescence analysis of phosphorylated signaling molecules, and we have performed multiplex qRT-PCR in order to evaluate the expression levels of genes downstream of Kit/Kitl. We have also included a new set of experiments aimed at evaluating macrophages in the earlier embryo in wild type and Kitl mutants, and have added new data focused on analyzing the HSC lineage in Kitl conditional knockouts. Moreover, we include a new set of quantifications of hematopoietic clusters, and other additional data and experiments described below. Following referees' comments, we have extensively rearranged and re-written the manuscript in order to better highlight the novel aspects of our work. We hope you will find our revised manuscript worthy of publication in EMBO reports.

Referee #1:

The manuscript by Azzoni et al. addresses the role of Stem Cell Factor (SCF), the established ligand of the cKit receptor tyrosine kinase, in hematopoiesis in the yolk sac and AGM. The authors cite data from the 1990s to make an argument that it is unclear whether SCF has important functions pre-fetal liver hematopoiesis. They establish that SCF is important, and endothelial cells constitute an important source of SCF. The authors conclude that SCF is a "critical regulator of HSPC development in the AGM and YS endothelial niche". HSPCs express cKit in all contexts studied - in vivo and in vitro. In an extremely well-established paradigm, SCF activates cKit to induce signaling that controls HSPC generation and function. The finding the SCF also has important functions pre-fetal liver hematopoiesis is not surprising, given the cKit⁺ hematopoietic precursors in this context. While it is useful to know that endothelial cells represent an important source of SCF at this stage, endothelial cells were known to secrete SCF. In summary, the current work appears to have been carefully executed, but the results only incrementally advance existing knowledge in the field. Furthermore, it is not evident that broader insights emerge from the studies.

“SCF activation of c-Kit to induce signaling that controls HSPC generation and function” is indeed a well-established paradigm in adult hematopoiesis. However, this paradigm has not been investigated *in vivo* in the AGM, yolk sac and early fetal liver hematopoietic niches. Indeed, very few known adult niche factors and cell components have been examined in the yolk sac and AGM niche *in vivo*. Data have mostly been obtained using *in vitro* cultures, which are not necessarily representative of the *in vivo* niche. Moreover, some factors were reported to show opposing effects in embryo and adult hematopoiesis (Gao et al., *Development* 2018; Mirshekar-Syahkal, *Stem Cells* 2014). More broadly, this underlines the need for experiments to test paradigms of adult hematopoiesis directly in the embryo, to provide the critical experimental foundations for further studies (Gao et al., *Development* 2018).

We examined the role of Kitl in the YS and AGM environments precisely because it is a strong candidate. By firmly establishing its requirement in embryonic hematopoiesis and identifying critical Kitl-generating cellular components in the yolk sac and AGM niche our work creates a solid foundation and new inroads to further explore the micro-environmental niche and other factors supporting the EMPs and the HSC lineage in the embryo. We have now extensively re-written the manuscript in order to focus on our new findings (see also our response to Referee #3). As detailed below, we have additionally performed new experiments to address the other comments of Referee #1.

Additional Comments:

1) Evidence was provided that SCF "has a previously unrecognized role in YS EMP proliferation". This conclusion is based on a ~2 fold change in proliferation. Does this involve a unique mechanism, or does it recapitulate known mechanisms by which SCF/cKit control proliferation? In addition, the magnitude of the change seems quite small. Is that because SCF is only a minor proliferative determinant?

EMPs are a heterogeneous population, consisting of clonal progenitors for the erythroid, myeloid and mixed myeloid/erythroid lineages. Of interest, in the absence of Kitl the biggest decrease in CFU-C was seen among the CFU-G/M/GM, with erythroid and mixed GEMM less severely affected. Kitl is known to synergize with other cytokines, such as EPO in erythroid cells (e.g. Wu et al., *PNAS* 1997), TPO in fetal hematopoietic stem/progenitor cell function (Antonchuk et al., *Blood* 2004) and, as recently shown, Oncostatin M in the expansion of HSCs/progenitors in zebrafish CHT (Mahony et al., *Stem Cell Reports* 2018). With respect to the magnitude of the change, the 50% decrease in proliferation is found reproducibly in the E9.5 to E11.5 yolk sac, using different techniques. This is accompanied by a 50% reduction in EMP and CFU-C numbers, and in our opinion represents a substantial decrease in cellular output, which we showed to have significant consequences for fetal liver hematopoiesis and tissue macrophages. Importantly, the changes in proliferation are reflected at the gene expression level, as discussed in our reply to point 2 below. We have added these new data in **Figure 8** and **Appendix Figure S3** and added a paragraph to the discussion (**page 12**).

2) SCF depletion impaired EMPs, tissue macrophages, LMPPs and HSCs. Does SCF function via identical mechanisms in these contexts, or are common SCF-derived signals interpreted differently? Providing new mechanistic insights would have potential to yield important results – results that might broadly inform SCF-cKit mechanisms.

Referee #1 raises an interesting question. We found that the lack of Kitl negatively impacted the proliferation of YS and FL EMPs, as well as erythroid progenitors (**Figures 1,2,3**), whereas proliferation was not significantly affected in Pre-HSC type II (**Figure EV4**). The effect of Kitl on Pre-HSCs seems instead to be linked to promoting their maturation and survival (**Figure 5**). As reported in the manuscript, we think that the effect on tissue macrophages is secondary to the decrease in EMP numbers, as tissue macrophages do not express Kit (Gomez-Perdiguero and Geissmann, *Nat Immunol* 2016; Mass et al., *Science* 2016). To address the referee's question we performed a new series of experiments to assess the mechanism downstream of Kit receptor activation in the YS, AGM and FL.

The signaling pathways downstream Kitl/Kit in hematopoiesis (and other tissues) have been well studied using a range of model systems, although they, to our knowledge, have not been assessed in embryonic hematopoietic cells. Well-described pathways are the phosphoinositide 3-kinase (PI3K) pathway converging on Akt, and the mitogen-activated protein kinase (MAPK) pathway, converging on Erk (reviewed in Kent, *Clin Cancer Res*, 2008; Roskoski, *Biochem Biophys Res Commun* 2005). To determine whether loss of Kitl affects the activation of either of these pathways (PI3K and MAPK) *in vivo*, we assessed phosphorylation of Akt and Erk in wild type and mutant embryos. Within the constraints imposed by working with primary embryo material we used whole-mount immunofluorescence for this (phospho-flow cytometry or biochemical assays are not feasible due to the number of cells required for these techniques and/or the rapid loss of phosphorylation which is incompatible with the collagenase digestion and staining times required for processing embryonic cells). As shown in the **new Figure 8A-D**, the PI3K and MAPK pathways were active in the AGM,

YS and FL and loss of *Kitl* resulted in reduced phosphorylation of Akt and Erk in aortic hematopoietic clusters and in YS EMPs. In FL *Kit*⁺ cells we did not detect changes in Akt or Erk phosphorylation, despite the decreased numbers of hematopoietic progenitors. Taken together, our data indicate that *Kitl* acts through, and is required for, the activation of the PI3K and MAPK pathways in aortic clusters and YS EMPs. In contrast, in FL *Kit*⁺ cells other niche factors may compensate for the activation of the PI3K and MAPK pathways, and other signaling pathways may be functionally relevant downstream of *Kitl* (e.g. Jak/Stat).

In a parallel approach, we examined the expression of selected genes reported downstream of *Kit/Kitl* by multiplex qRT-PCR in YS EMPs, FL EMPs, FL erythroid progenitors, LMPPs, and AGM Pre-HSCs isolated from wild type and *Sl/Sl* embryos (**new Appendix Tables S4 and S5**). We observed a strong upregulation of the anti-proliferative gene *BTG1* in *Sl/Sl* YS EMPs and, to a lesser extent, in FL erythroid progenitors (**new Figure 8E**). This indicates that *Kitl* represses *BTG1*, possibly through PI3K/Akt and FOXO3A, to control EMP proliferation *in vivo*, similar to what was previously described in erythroid cell lines (Bakker et al., *J Cell Biol* 2004). Accordingly, we observed that FOXO3A is expressed in YS EMPs (see figure below).

Immunofluorescence for FOXO3A on frozen sections of E10.5 wild type embryos.

Of note, expression levels of *BTG1* were normal in Pre-HSCs, suggesting that this pathway is specific to EMPs. Along with the changes in *BTG1*, cyclin-dependent kinase inhibitors 1a and b were both upregulated in YS EMPs, supporting a role for *Kitl* in suppressing cell cycle inhibitors in order to promote proliferation of these cells. *Myc*, which mediated cell cycle progression in erythroblasts (Munugalavada, *Mol Cell Biol* 2005) and *Stat5a*, known to promote erythroid proliferation and differentiation downstream of *Kit* (Kapur, *J Biol Chem* 2001), were both decreased specifically in *Sl/Sl* FL erythroid progenitors. However, other cell cycle genes were generally unaffected in this population (**new Appendix Figure S3**). In the AGM, *Sl/Sl* Pre-HSCs showed specific deregulation of genes involved in apoptosis such as increased expression of pro-apoptotic gene *Bax* and decrease in the anti-apoptotic gene *Bcl2* (**new Figure 5J, Appendix Figure S3**), in line with the increased apoptosis in *Sl/Sl* Pre-HSC type II detected by Annexin V. No widespread changes in cell cycle-associated genes were seen. Taken together, these results indicate that the primary signaling pathways downstream of *Kit/Kitl* are likely to be the same, or similar, in the different cell populations analysed in YS and AGM, but the cell-specific functions elicited by *Kitl* signaling are dependent on the cellular context. These data are included in the revised manuscript.

3) *In the yolk sac, is Tie2-CRE only active in endothelial cells? Do endothelial cells generate the majority of the SCF in the yolk sac, or is endothelial cell-derived SCF uniquely important?*

In the E8.5 yolk sac, *Tie2-Cre* recombines in the majority of endothelial cells, but not/very little in other cell types such as visceral endoderm or hematopoietic cells (mainly primitive erythrocytes in the E8.5 yolk sac). See Kisanuki et al., *Developmental Biology* 2001, Figure 4H:

X-Gal staining of E8.5 Tie2-Cre;CAG-CAT-Z double transgenic embryos, showing the yolk sac blood islands (bi). Endothelial cells are clearly labelled, while virtually all blood cells (primitive E) are unlabelled.

From: Figure 4H, Kisanuki et al., *Developmental Biology* 2001

Concerning the expression of *Kitl* in the yolk sac, apart from expression in yolk sac endothelial cells (arrowheads in **Figure 6C** and **Figure 6D**), we also detected *Kitl*-Tomato expression in ‘mesenchymal’ (flow cytometry; **Figure EV5D**) and perivascular (whole-mount immunofluorescence; arrow in **Figure 6C**) cells. Despite the fact that endothelial cells constitute only 4-5% of all *Kitl*-expressing cells in the E10.5 yolk sac (as detected by flow cytometry, data not shown), our functional data indicate that endothelial-derived *Kitl* is particularly important in the yolk sac as the deletion in *Tie2* expressing cells phenocopies the effect on EMPs in the *Sl/Sl* yolk sac (**Figure 7B,C**). This does not exclude a role for *Kitl* derived from other cell types, but makes it unlikely to be a large role. In contrast, in the fetal liver we found that deletion of *Kitl* in *Tie2*⁺ endothelial cells did not affect EMPs (**Figure 7B**), indicating there is a different niche for EMPs in this tissue. HSCs, on the other hand, seem to be dependent on endothelial *Kitl* throughout their development in the AGM (**Figure 7B,D**; **Figure EV4D**) and FL (**new data added in Figure 7F and Figure EV4**).

Referee #2:

This work investigates the importance of Kit-ligand (Kitl) in early embryonic hematopoiesis. The authors found that alteration of Kitl strongly reduces the population of erythroid-myeloid-progenitors originating in the YS and of the pre-HSCs in the AGM. They used three transgenic mouse lines to identify the specific expression of Kitl in the embryonic tissues and study alteration of the Kitl-dependent pathway using broad and endothelial-specific ablation.

General Comments:

The main question of the authors is of value for the field, as the factors controlling early embryonic hematopoiesis are largely unknown. Importantly, they base their analysis largely on in-vivo systems and genetic mutants. On the other hand, previous reports already defined the importance of Kitl in the AGM, in vitro (Rybtsov et al. 2014 and other previous publications) and more recently in vivo (Souilho et al 2016). This partially hampers the level of novelty of the findings here described, but the authors come to slightly different conclusions and in addition focus on the YS EMPs and primitive macrophages, which has been less intensively studied.

The recent study by Souilhol et al., *Nature Communications* 2016 reported on *Kitl* expression in the AGM (qRT-PCR and whole-mount imaging), but did not assess what cell types express *Kitl*, or what is the requirement for *Kitl* in *in vivo* HSC development. The Souilhol experiments assessed the effect of *Kitl* in an *ex vivo* culture system, the results of which indeed differ from our data in the embryo, underlining the need to analyze the role of micro-environmental factors using *in vivo* approaches (Gao et al., *Development* 2018).

The paper will gain by important revisions as described in more details below.

1) The authors found that alteration of Kitl results in decreased number of macrophages in the majority of tissues and decreased number of cKit⁺ cells in the hematopoietic clusters of the AGM. It is possible that the two phenotypes are linked to each other, as the macrophages are present in the

surrounding of the hematopoietic clusters in the AGM (Travnickova et al, 2015) and might contribute to the maintenance of their structure and functionality. Can the authors test whether macrophages are equally present in clusters in the presence or absence of kitl?

This is an interesting point. To address this possibility we have performed whole-mount imaging of macrophages in the wild type and *Sl/Sl* AGM at E10.5, and flow cytometry analysis of macrophages in E10.5 and E11.5 tissues. These new data are included in **Figure 4 and Appendix Figure S1**. Interestingly, at both E10.5 and E11.5 we detected a significant decrease in both percentage and absolute numbers of macrophages in the AGM+VU (**Figure 4E-F**). However, when we quantified the number of macrophages within, or in close proximity to, hematopoietic clusters in the aorta using whole-mount imaging, we saw no difference between wild type and *Sl/Sl* (**Figure 4D**). Moreover, the number of macrophages in/surrounding the clusters was rather low (approx. 15 per aorta on average) compared to the number of macrophages in the whole AGM (in the wild type, approx. 600 at E10.5, and 3000 at E11.5). Therefore, although we cannot formally rule out the possibility that the overall decrease in AGM macrophages contributes to (part of the) hematopoietic phenotypes observed in the *Sl/Sl* AGM, the fact that macrophages in (the vicinity of) the clusters are present in normal numbers suggests that the pre-HSC defect in the AGM is a direct consequence of the lack of Kitl. These considerations are added to the discussion (**page 11**).

Of note, similarly to the referee's argument, we considered the possibility that a decrease in fetal liver macrophages in erythroblastic islands could contribute to the defect in erythroid maturation in the fetal liver. However, since we observed no significant difference in the number of macrophages in the early FL (E10.5/E11.5; **Appendix Figure S1**), we conclude this is unlikely to contribute to the early erythroid defects we report here.

2) The phenotype of the Sl/Sl mice in the AGM reflects in decreased number of cKit+ cells and decreased reconstitution potential of the AGM-derived HSCs. Have the authors counted the number of clusters? In the absence of kitl, are Kit+ cells reduced because clusters are smaller or there are less clusters or both? Is there more EHT, so more clusters of more individual kit+ cells in the endothelium?

Hematopoietic clusters in the *Sl/Sl* dorsal aorta appeared smaller (**Figure 6A**). To quantify this observation, we measured the number, size and position of clusters in the wild type and *Sl/Sl* aorta, similarly to the method applied by Yokomizo and Dzierzak, *Development* 2010. These new data are included in **Figure 5**. We detected a decrease of the total number of Kit+ cell clusters (>1 cell) attached to the aortic wall, from 88.4 ± 19.7 (n=7) in the wild type aorta to 65.6 ± 14.4 (n=7) in the *Sl/Sl* aorta (p=0.03). Large clusters (>10 cells) associated with the ventral wall of the aorta were specifically decreased (**Figure 5B**), reflecting the smaller size of clusters apparent in the whole-mount imaging. Smaller clusters or single Kit+ cells attached to the ventral wall were unaffected, indicating that EHT is not affected in *Sl/Sl* mutants. This is in line with the normal presence of pre-HSC type I in the *Sl/Sl* AGM (**Figure 5C-D**), which also indicated that EHT was unaffected and that the defect occurred well after the HSC lineage diverged from the HE. Note that EHT was also unaffected in the yolk sac, as EMPs initially emerged in normal numbers and were only affected at the stage of their expansion. Kit+ cells in the lumen of the aorta were also decreased (**Figure 5B**). This likely reflects (among others) the decrease in circulating EMPs and LMPPs (cf. **Figure 3, Figure EV1 and EV2**). Interestingly, lack of Kitl did not affect cluster size or number at the dorsal aspect of the aorta (**Figure 5B**), consistent with the absence of dorsal Kitl expression (cf. **Figure 6E**).

3) Is the lack of Kitl directly blocking the proliferation of functional HSCs or is it disturbing the maturation of the hematopoietic clusters as published by the Medvinsky lab? This may be difficult to investigate, but the authors should speculate on that.

This is an interesting point. As beautifully demonstrated by the Medvinsky lab, expansion of the AGM HSC lineage occurs mainly at the level of the pre-HSCs (Rybtsov et al., *Development* 2016). More recently, using Fucci mice, they showed that pre-HSCs proliferate more than dHSCs, suggesting that the expansion of the lineage is due to pre-HSC proliferation rather than continued *de novo* generation of these cells (Batsivari et al., *Stem Cell Reports* 2017). Pre-HSC I proliferation

dramatically slows over time, but the reason for this is unknown. Similarly, the role of proliferation in the maturation of the HSC lineage remains unknown.

We did not detect significant differences in the cell cycle status of pre-HSCs/HSC (**Figure EV4C**) in wild type versus *Sl/Sl* embryos, suggesting that *Kitl* does not affect proliferation during the generation of these cells. Of note, the staining for pHH3 in the aortic clusters is not specific to (pre-)HSCs, which could reconcile the BrdU data with the reduced pHH3 staining among the total *Sl/Sl* aortic cluster population. We did detect an increase in apoptosis in *Sl/Sl* pre-HSC type II/HSCs (**Figure 5I**), along with increased expression of the pro-apoptotic gene *Bax* and decreased expression of the anti-apoptotic gene *Bcl2* (**Figure 5J**). This indicates that promoting survival could be how *Kitl* affects pre-HSC numbers *in vivo*. We have incorporated these considerations in the revised discussion (**page 12**).

Minor points:

*Figure 1: Whole mounts with *kitl*-tomato are not clear. The authors choose to show the maximal projection of the embryo, but maybe a single stack image would be clearer. In this sense: In Fig 1A, is not clear to me what they want to show. Can the authors draw a line showing the conceptus? Is the YS mainly negative for *kitl*-Tomato? Figure 1C shows that there is not.*

Based on the referee's feedback, we have restructured **Figure 1**, which is now **Figure 6** in the revised manuscript. We have incorporated all of the *Kitl*-tomato expression data in **Figure 6** and in related **Figure EV5**. We kept the maximum intensity projection of the E8.5 and E10.5 embryos in the main figure, as we believe that this overview provides important spatial context. However, we have added lines outlining the edges of the embryo proper and yolk sac regions within the conceptus. We have also added boxed areas in **Figure 6A**, of which single slice images are shown at higher magnification (**Figure 6B-C**). The image in **Figure 6B**, a higher magnification of the para-aortic splanchnopleura region, shows that *Kitl* is expressed in endothelial cells of the paired aortae already at E8.5. Within the YS, overall *Kitl*-tomato expression is lower than in the embryo proper (**Figure 6A**), with individual endothelial and perivascular cells expressing *Kitl*-tomato (**compare Figure 6C with Figure EV5C**). The resolution of the whole-mount in **Figure 6A** does not allow detection of these cells; the single slice panel in **Figure 6C** shows these cells more clearly. *Kitl*⁺ endothelial cells in the YS are also detected at later stages. To make this clearer, we now show the E10.5 panel in **Figure 6E** at a higher magnification. We now also specify what the arrows are indicating (apologies for the original omission).

*Fig 1B: Whole mount imaging of the *Kitl*-Tomato transgene should be improved or supported by additional magnification of the aortic region or showing a specific stack.*

We cannot further improve the quality of the image itself, which to the best of our knowledge represents the state-of-the-art. However, in order to show *Kitl* expression in the aortic region in more detail, we now included a close-up of this region in **Figure 6E** (area now indicated by a box), which is a single z-stack slice image from the same embryo. We hope the referee finds this helpful.

Fig 1D.

*The claim that perivascular cells are expressing *kitl* is not well supported by the staining of SMA because at 10.5 the perivascular structures are not well defined and SMA is expressed mainly by endothelial cells. In fact the authors will conclude that *kitl* from endothelial cells is responsible for the hematopoietic effects, which makes this part more confusing. The authors should consider restructuring or remaking figure 1 for clarity.*

As can be seen from the whole-mount now in **Figure 6E**, there is extensive *Kitl*-tomato expression in the sub-aortic mesenchyme. We believe that part of these are smooth muscle cells. The referee is right that perivascular structures are not well defined yet at E10.5. However, SMA⁺ cells begin to populate the region of the aorta already at E9/E10 (Wasteson et al., *Development* 2008). These take a few days to develop into the smooth muscle layers seen in the E14.5 and adult dorsal aorta, but the first of these are present already at E10.5. In the original submission the VE-Cad staining was not very strong, so to better indicate the staining of SMA in non-endothelial cells we have now repeated the SMA staining with CD31 and included the new image in **Figure 6F**. We trust it is now clearer that there are several SMA⁺ CD31⁻ *Kitl*-tomato⁺ cells located a few cell diameters from the lumen,

representing these first perivascular smooth muscle cells. It should be noted that the expression of Kitl in the AGM extends beyond these cells, and as the referee comments, Kitl expression does not equal functional relevance of these cell types in the hematopoietic niche.

Sup Fig1

SF1A: No clear positive populations for kitl-tomato are shown in the dotplots for E8.5, AGM+VU or YS. Could the authors do double staining with kitl antibody and check the correlation of expression? That would be really informative.

We have tried staining sections with Kitl antibody. The antibody did not work very well (we tried different fixatives, etc.), but more importantly, the staining would have been difficult to interpret as the endogenous Kitl protein can also be secreted, while the transgenic Kitl-tomato reporter is restricted to the cells that produce Kitl. The Kitl-tomato transgene was shown to faithfully report Kitl expression by the group of Claus Nerlov (Buono et al., *Nature Cell Biology* 2016). Since they previously assayed adult tissues, we also assessed the faithfulness of the reporter in our embryonic tissues of interest. qRT-PCR for Kitl in tomato+ and tomato- embryonic cell populations showed again good correlation. Tomato expression is indeed a continuum, which is not surprising given the widespread expression of the tomato in various cell types. We have included here the FACS plots of wild type embryos to allow assessment of our gating strategy.

SF1B: The label on the Y-axis is unclear.

The Y-axis label was the same for all plots, but in order to make this clearer we have now labeled all plots individually (**Figure EV5B**).

SF1D and E: The authors exclude MC with CD41 and CD45 staining. Can they show the dotplots? Are these markers used in the definition of HC or HP?

EC and MC were analyzed as Ter119- VE-Cad+ Kit- CD41- CD45- and Ter119- VE-Cad- Kit- CD41- CD45- cells, respectively, as indicated in the figure. The dot plots for CD41 and CD45 expression within the EC and MC gates (of the VE-Cad/Kit plot) are shown below and are now included in Figure **EV5E-F** for clarity. Kitl-tomato expression is shown in the final gated populations. CD41 and CD45 were not used for the analysis of HC or HP as those were identified here as Ter119- VE-Cad+ Kit+ and Ter119- VE-Cad- Kit+, respectively. Including CD41 or CD45 in the gating strategy for Kit+ HC and HP in the yolk sac would not change the percentages as neither of these populations show significant Kitl-tomato expression.

Flow cytometric plots relative to Figures EV5E and F, showing CD41 and CD45 expression within EC (Ter119- VE-Cad⁺ Kit-) and MC (Ter119- VE-Cad- Kit-) subsets.

Fig2F: The FACS plot does not show any cell in the M phase in the dot plots, although several cells express the mitotic marker PHH3 in Fig2E. Can they reconcile both findings?

This was due to the low number of events we could record for this cell population. We did observe cells in G2/M-phase in other tissues where we could collect more events (e.g. fetal liver, see figure below).

Flow cytometric cell cycle analysis of wild type and *SI/SI* E11.5 FL EMPs (Ter119- Kit⁺ CD41⁺ CD16/32⁺).

We have now repeated the BrdU incorporation experiment with more embryos, allowing us to record more events for YS EMPs. The YS plots in **Figure 1F** (previously Figure 2) now show cells in the M phase for wild type embryos, and much less so in *SI/SI* embryos.

Figure6

Fig6B: Quantification lacking for apoptosis

Quantification of apoptosis on the whole-mount images does not allow quantification in pre-HSC I and II/HSC subsets. To quantify apoptotic cells in these populations, we previously chose to perform Annexin V staining by flow cytometry. We have now performed whole-mount immunofluorescence staining of cleaved caspase 3 on more embryos in order to more accurately quantify the number of apoptotic cells in the aorta. We report this quantification in **Figure 5H**, which shows a significant increase in the number of apoptotic Kit⁺ cells in the aorta of *SI/SI* embryos.

Figure

Fig7A,B: Are CD45⁺ cells still present in the cluster after Kit1 ablation?

Yes, cells with the phenotype of cluster cells (VE-Cadherin⁺ TER119-CD41⁺CD45⁺) are still present in the *SI/SI* AGM, as detected by flow cytometry. We also detected CD45⁺ cells within aortic hematopoietic clusters of *SI/SI* embryos by whole-mount immunofluorescence (see figure below).

Whole-mount immunofluorescence of E10.5 *SI/SI* embryos.

Referee #3:

EMBO 45477-T "A critical role for Kit ligand in mouse pre-liver hematopoiesis" by Report-2017-de Azzoni and colleagues. The authors studied expression and function of Kit ligand (Kitl) in embryonic and fetal hematopoiesis. They argue initially that little is known about extrinsic factors regulating the hematopoietic system, yet in this paper the author then go on and study one of the best known and most intensively studied hematopoietic growth factors, Kitl. Along these lines, the argument that factors may help in supporting hematopoietic stem cell derivation or expansion in vitro is perhaps ok but from all we know it is clear that Kitl is not a factor capable of doing this. Also on the critical side, Kit expression has long been studied in yolk sac and subsequent stages (see for example the detailed report by Matsui, Zsebo & Hogan. Embryonic expression of a hematopoietic growth factor encoded by the Sl locus and the ligand for c-kit. Nature 1990).

Apologies for not being clearer. What we meant to say is that little is known about extrinsic factors regulating *the initial generation* of hematopoietic stem and progenitor cells in the embryo. This is not at odds with studying Kitl, as it is a strong candidate player given its critical role in the adult, where it is in fact known to promote adult HSC maintenance and expansion *in vitro* (see work from the Connie Eaves lab, e.g. Petzer et al., *J Exp Med*, 1996; Kent et al., *Blood* 2008; Knapp et al., *Stem Cell Reports* 2017 among others).

We appreciate that the role for Kitl in adult HSCs is extremely well studied, but its role in the generation of HSCs and progenitor cell the YS and AGM *in vivo* is not. The studies from the 1990s concluded that neither HSCs (Ikuta and Weissman, 1992, "Evidence that HSCs express c-kit but do not depend on steel factor for their generation") nor progenitor cells from the early yolk sac and fetal liver (Ogawa et al., 1993, "Expression and function of c-Kit in fetal hemopoietic progenitor cells: transition from the early c-Kit-independent to the late c-Kit-dependent wave of hemopoiesis in the murine embryo") depend on SCF. More recently reports by the Medvinsky group indicated that SCF is required for HSC maturation *in vitro* (Rybtsov et al., *Stem Cell Reports* 2014, and additional papers from that group), though they did not assess this *in vivo*, nor the role for SCF in progenitors of the early yolk sac and fetal liver. We believe that our work is of merit as it directly and comprehensively demonstrates the requirement for Kitl *in vivo* in YS-derived progenitor cell types, including early FL erythroid progenitors and fetal tissue macrophages, and supports a direct requirement for Kitl in the emerging AGM HSC lineage *in vivo*. The apparent discrepancy between our findings and the studies from the 1990s can be explained by the tissue, time point and cell type analysed (Ikuta and Weissman) and the experimental approach used (Kit blocking Ab); of note, neither study assessed primary AGM HSCs. Our findings underline the importance of studying niche factors *in vivo* in their relevant hematopoietic niches using genetic models (Gao et al., *Development* 2018).

As to the expression of Kitl in the yolk sac, Hogan and colleagues analyzed this by in situ hybridization and Northern blot (Nature 1990). The resolution of these analyses did not allow the identification of the cell types that express Kitl in the yolk sac, which we assessed here using a Kitl-tTomato reporter mouse line. We added the Hogan paper to our citations.

On the other hand, one could argue that the authors have re-addressed expression and function of Kitl in embryonic and fetal hematopoiesis, and today, i.e. 25 years later, more sophisticated tools are available and have been utilized by the authors here (e.g. genetic reporter). The data include visualization of Kitl expression in a fluorescent reporter, careful and quantitative re-analysis of hematopoiesis in natural Sl (Steel; lacking both membrane bound and soluble Kitl) mutant embryos and fetuses, and endothelial cell-restricted conditional deletion of Kitl. These experiments are well done and the data seem solid.

To our knowledge, our study is the only comprehensive and direct analysis of the requirement for Kitl in well-defined hematopoietic stem and progenitor cells and their niches in the embryo and fetus. We are pleased that the referee judges the quality of our work positively.

In summary, the paper is written to suggest that very little was known about Kitl expression and function. This is exaggerated. While much of the data look like details in our overall picture in the Kit-Kitl system, the data still seem worth publishing, for as long as a revised paper focuses on the

specific and new aspects shown here (and avoids construed justifications for why all of this was studied).

This point is well taken. As the referee suggested, we have extensively rewritten the paper in order to put more emphasis on our new findings in embryonic and fetal hematopoiesis. We showed a previously unrecognized requirement for Kitl in YS and FL EMPs and their progeny, such as erythroid cells and tissue macrophages, and our data support a requirement for Kitl specifically at the level of AGM pre-HSC II/HSCs. In addition, we showed that the HSC lineage remains dependent on endothelial Kitl throughout its development in the AGM and FL niches. In contrast, EMPs only depend on endothelial Kitl in the YS but not the FL. Finally, new data on signaling and gene expression downstream of Kitl/Kit signaling (**Figure 8, Appendix Figure S3**) showed that the interpretation of this signal is dependent on cell type/tissue, and provided further support for regulating expansion/proliferation of EMPs in the YS and FL, versus maturation and survival of (pre-)HSCs in the AGM. We believe that the restructuring of the manuscript and addition of new experimental data has improved the clarity and added to the novelty of our message.

Minor: Formally speaking, the current nomenclature is Kit not c-Kit

We are aware of this, but chose to use 'c-Kit' to better contrast with 'Kitl' in figures, as we had noted that in data presentations 'Kit' vs 'Kitl' could at times be hard to distinguish by the reader. We have now changed 'c-Kit' into 'Kit' throughout the revised manuscript.

2nd Editorial Decision

20th Jul

Thank you for the submission of your revised manuscript to our editorial offices. We have now received the reports from the two referees that were asked to re-evaluate your study (you will find enclosed below). As you will see, both referees now support the publication of your manuscript in EMBO reports.

Referee #1:

The manuscript has been extensively revised, and the revisions appropriately address my prior recommendations. Various effects are quantitatively small, e.g. the gene expression data in Fig. 8, calling into question the importance of the respective mechanisms. However, overall, the work will likely be of interest to investigators in this specific field.

Referee #2:

Authors have addressed all concerns and the manuscript is greatly improved.

Corresponding Author Name: Marella F.T.R. De Bruijn

Manuscript Number: EMBOR-2017-45477V2